# Exogenous putrescine attenuates the negative impact of drought stress by modulating physio-biochemical traits and gene expression in sugar beet (*Beta vulgaris* L.)

Md Jahirul Islam[1,2], Md Jalal Uddin[3,4], Mohammad Anwar Hossain[5]*, Robert Henry[6], Mst. Kohinoor Begum[2], Md. Abu Taher Sohel[7], Masuma Akter Mou[8], Juhee Ahn[3], Eun Ju Cheong[9], Young-Seok Lim[1]*

1 Department of Bio-Health Convergence, College of Biomedical Science, Kangwon National University, Chuncheon, Republic of Korea, 2 Physiology and Sugar Chemistry Division, Bangladesh Sugarcrop Research institute, Ishurdi, Pabna, Bangladesh, 3 Department of Medical Biomaterials Engineering, College of Biomedical Science, Kangwon National University, Chuncheon, Republic of Korea, 4 Research Group for Host-Microbe Interactions, Department of Medical Biology and Centre for New Antibacterial Strategies (CANS), UiT—The Arctic University of Norway, Tromsø, Norway, 5 Department of Genetics and Plant Breeding, Bangladesh Agricultural University, Mymensingh, Bangladesh, 6 Queensland Alliance for Agriculture and Food Innovation, University of Queensland, Brisbane, Qld, Australia, 7 Agronomy and Farming System Division, Bangladesh Sugarcrop Research Institute, Ishurdi, Pabna, Bangladesh, 8 Department of Agronomy, Bangladesh Agricultural University, Mymensingh, Bangladesh, 9 Division of Forest Science, College of Forest and Environmental Sciences, Kangwon National University, Chuncheon, Korea

* anwargpb@bau.edu.bd (MAH); potatoschool@kangwon.ac.kr (YSL)

## Abstract

Drought tolerance is a complex trait controlled by many metabolic pathways and genes and identifying a solution to increase the resilience of plants to drought stress is one of the grand challenges in plant biology. This study provided compelling evidence of increased drought stress tolerance in two sugar beet genotypes when treated with exogenous putrescine (Put) at the seedling stage. Morpho-physiological and biochemical traits and gene expression were assessed in thirty-day-old sugar beet seedlings subjected to drought stress with or without Put (0.3, 0.6, and 0.9 mM) application. Sugar beet plants exposed to drought stress exhibited a significant decline in growth and development as evidenced by root and shoot growth characteristics, photosynthetic pigments, antioxidant enzyme activities, and gene expression. Drought stress resulted in a sharp increase in hydrogen peroxide ($H_2O_2$) (89.4 and 118% in SBT-010 and BSRI Sugar beet 2, respectively) and malondialdehyde (MDA) (35.6 and 27.1% in SBT-010 and BSRI Sugar beet 2, respectively). These changes were strongly linked to growth retardation as evidenced by principal component analysis (PCA) and heatmap clustering. Importantly, Put-sprayed plants suffered from less oxidative stress as indicated by lower $H_2O_2$ and MDA accumulation. They better regulated the physiological processes supporting growth, dry matter accumulation, photosynthetic pigmentation and gas exchange, relative water content; modulated biochemical changes including proline, total soluble carbohydrate, total soluble sugar, and ascorbic acid; and enhanced the activities of antioxidant enzymes and gene expression. PCA results strongly suggested that Put

**Data Availability Statement:** All relevant data are within the paper and Supporting information file.

**Funding:** The author(s) received no specific funding for this work.

**Competing interests:** The authors have declared that no competing interests exist.

conferred drought tolerance mostly by enhancing antioxidant enzymes activities that regulated homeostasis of reactive oxygen species. These findings collectively provide an important illustration of the use of Put in modulating drought tolerance in sugar beet plants.

## Introduction

Abiotic stresses created by a range of hostile environments are considered a prime limiting factor for plant growth, development, and productivity worldwide [1, 2]. Due to their sessile behavior, plants need to develop intrinsic metabolic capabilities to cope with harsh conditions by evolving cellular, physiological, and morphological defense mechanisms under stress conditions [2–4]. Drought is considered the most restrictive factor among the various abiotic stresses that significantly impact the plant life cycle [5]. Due to global climate change, the frequency and duration of water stress or drought will continue to increase in the future, and we need to develop drought-tolerant varieties or management techniques for each crop species.

Sugar beet (*Beta vulgaris* L.) is well-known as the second most important sugar-producing crop after sugarcane, contributing to about 40% of world sugar production [6]. Sugar beet production depends on many factors, but drought is considered the critical limitation for reducing sugar beet yield from 5 to 30% [7, 8]. Drought has a detrimental effect on crop yield through its direct negative impact on plant growth and development, pigment production, photosynthetic rate, nutrient accumulation, and osmotic adjustment [9, 10]. Water scarcity hampers crop yield by affecting plant growth, development, and quality [5, 11]. Plants grown under drought stress produce a pool of reactive oxygen species (ROS) through physiological, biochemical, morphological, and molecular changes [5], causing an imbalance in component quantities and dysfunction of their typical defensive system [12]. This interruption of the defensive system provokes the overproduction of ROS consisting of both non-radical (hydrogen peroxide, $H_2O_2$) and free radical species (superoxide, $O_2^{-\bullet}$; hydroxyl radical, $OH^\bullet$) that are highly detrimental to plant cells [13].

Excess production of ROS may cause damage to the cell, reduce enzymatic activity, degrade protein and nucleic acid, and finally result in cell death [14, 15]. Therefore, a ROS detoxification system needs to be activated in the plant cells as their defense mechanism to minimize the toxic effects [16]. Excessive ROS accumulation is subject to control by a complex mechanism of enzymatic and non-enzymatic antioxidant systems. These enzymatic antioxidant elements mainly include superoxide dismutase (SOD), catalase (CAT), ascorbate peroxidase (APX), and guaiacol peroxidase (GPX), present in different subcellular organelles of the plant [5, 14]. A positive correlation between antioxidant enzyme activity and their gene expression has been reported in some previous studies [17, 18]. Recently, studies of the regulation of gene expression have been identified as essential to improve the understanding of abiotic stress tolerance in plants [17, 19]. It is plausible that over-expression of genes related to antioxidant enzymes may enhance abiotic stress tolerance in plants.

Polyamines (PAs) are aliphatic polycations that contain two or more amino groups. PAs can respond to different abiotic and biotic stresses by regulating physiological processes [20, 21]. These responses might be due to the ability of PAs to adjust osmosis and detoxify the cell by scavenging ROS [22, 23]. Drought affects morpho-physiological processes and inhibits the antioxidant enzyme activities of the plant. Exogenous PA application can effectively upregulate the situation; among different types of PAs, putrescine (Put) was found to have the greatest significance in lettuce seedlings [24]. In the PA biosynthesis pathway, Put is the central product

containing two amino groups and acts as a synthetic precursor of spermidine and spermine. Among the three different biosynthetic routes, primary Put synthesis follows the removal of nitrogen atoms from agmatine (Agm) to form N-carbamoyl Put (NCPA). After that, NCPA is hydrolyzed by N-carbamoylputrescine amidohydrolase (NCPAH) to form Put [21].

Polyamines act as universal organic polycations and are associated with a wide range of the plant's fundamental processes like growth and development, senescence, and notably adaptation to abiotic and biotic stresses [25]. Putrescine can bring change in the plasma membrane of guard cells by regulating the size of the potassium channel pores to control the pore opening and closing, thereby preventing water loss in the plant [21, 26]. Controlled foliar application of Put can trigger some physiological processes and induce osmotic adjustment molecules like proline, total soluble sugars, and amino acids in plants [21]. Recently it was reported that the catabolic activity of Put compensated for total chlorophyll and chlorophyll fluorescence from saline stressed ginseng seedlings, thus protecting the plants from stress-derived damage and restoring the morpho-physiological activities [27]. It has also been reported that PAs affect DNA, RNA, and protein biosynthesis, exacerbates plant growth and development, slows aging, and protect the membrane from oxidative damage by removing free radicals in plants [6]. The beneficial roles of exogenous Put conferring stress tolerance in some plants have been well documented [21, 28–30]; however, many aspects of Put-mediated drought stress tolerance remain elusive. Here we provide the first report of the potential beneficial roles of exogenous Put in modulating drought stress tolerance in sugar beet by regulating physio-biochemical traits and gene expression.

## Materials and methods

### Plant material and experimental design

Sugar beet seedlings were grown in a semi-controlled greenhouse in the department of Bio-Health Convergence, Kangwon National University, Chuncheon, Korea. The environmental conditions such as temperature, relative humidity (RH) and average photoperiod were recorded at 30/25˚C (day/night), 60–70% RH and 12 h respectively. Seeds of sugar beet were provided by Asia Seed Co., Ltd, Seoul, Korea, (**SBT-010**) and Bangladesh Sugarcrop Research Institute (BSRI), Bangladesh (**BSRI Sugar beet 2**). The seeds of 2 genotypes were sterilized by treatment with 70% (v/v) ethanol (2 min) followed by 0.1% (w/v) $HgCl_2$ (8 min) and 0.2% (w/v) thiram (20 min). After that, the sterilized seeds were placed in a16 cell plug tray (27 × 27 × 6 cm) containing commercial horticultural soil (Bio-soil No. 1, Heungnong Agricultural Materials Mart, Korea). The seedlings were irrigated daily using tap water to the field capacity level for up to 15 days. After that, seedlings were transferred to a growing pot (10 cm × 9.5 cm) containing 180-gram of a formulated substrate (commercial horticultural soil: organic manure = 3:1) and were irrigated daily using tap water to the field capacity level. At the thirty-day-old stage, the seedlings were subjected to drought stress at 30% field capacity (FC) for ten days, while the control plants were grown at 90% FC. After that, the seedlings under drought stress were treated with Put (1,4-diaminobutane; Tokyo Chemical Industry Co., Ltd., Japan) at concentrations of 0.3 mM, 0.6 mM, and 0.9 mM, while control plants were treated with water only. Put, and water was applied to both sides of the leaves of seedlings for one time with 1% Tween-20 (v/v). After ten days of treatment, the plant samples were randomly collected for further analysis.

### Determination of plant growth parameters and leaf relative water content (LRWC)

Plants were sampled from 6 pots selected at random for data collection. The plant growth rate was calculated based on height differences between 1st and 10th days of treatment application.

On the 10th day from the onset of treatments, seedlings were placed in an oven at 60°C for 48 h, to determine the dry weight. Leaf area was measured according to the formula A = W × L × 0.675 [31], and leaf relative water content was measured according to [5] using the formula RWC (%) = [(FW- DW)/(TW-DW)] × 100.

## Determination of photosynthetic pigments, photosynthetic traits, and photosynthetic fluorescence parameters

Chlorophyll *a*, chlorophyll *b*, and carotenoids of freeze-dried leaf samples were determined according to the method described by Lichtenthaler [32]. The photosynthetic fluorescence parameters were measured using a Fluor Pen FP 100 (Photon system Instruments, Czech Republic) by measuring transient OJIP under dark-adapted conditions for 20 min.

For gas exchange characteristics, Net photosynthesis rate (A, $\mu mol\ m^{-2}\ s^{-1}$), transpiration rate (E, $mmol\ m^{-2}\ s^{-1}$), and stomatal conductance ($g_s$, $mmol\ m^{-2}\ s^{-1}$) were measured using an ADC BioScientific LCpro gas analyzer. Fully expanded leaves at the third stem node from the top of each plant were chosen from 6 seedlings randomly selected from each treatment. The level of A, $g_s$, E, and water use efficiency (WUE) was measured at the ambient environmental conditions. The measurements of gas exchange were carried out in the middle of the day between 10 a.m. and 3.00 p.m. The measurements were made on the second leaf of each of six randomly selected seedlings. The photosynthetic WUE was calculated as the ratio A/E.

## Determination of malondialdehyde and $H_2O_2$ content

The fresh leaf samples (250 mg) were macerated in 0.1% trichloroacetic acid (5 mL). The homogenate was centrifuged at 12000×g for 10 min at 4°C, and the supernatant was stored at 4°C for analysis. Lipid peroxidation was determined by estimating the malondialdehyde (MDA) content in the leaves of the sugar beet seedlings. The procedure for estimation of MDA and $H_2O_2$ was described in our previously published article [5].

## Analysis of free proline content

Approximately 25 mg of freeze-dried plant material was used to estimate proline concentration. The concentration of proline was determined by following the method described by Bates [33]. The absorbance was taken at 520 nm with a UV-Vis spectrophotometer (UV-1800 240 V, Shimadzu Corporation, Kyoto, Japan), and calculations used an appropriate proline standard curve.

## Analysis of glycine betaine and mannitol content

A freeze-dried sample (25 mg) was extracted with 5.0 mL ethanol (80%) and sonicated for 1 hour at room temperature. The extract was then centrifuged at 5000 rpm for 10 min, and the supernatant was collected. The residue was re-extracted with a fresh aliquot of 5.0 mL ethanol (80%). The supernatants were combined, filtered through a syringe filter (0.45 μM, Millipore, Bedford, MA, USA), and stored at 4°C for analysis.

An HPLC system (CBM 20A, Shimadzu Co, Ltd., Kyoto, Japan) with a 5 μm C18 column (25 cm × 4.6 mm) and ELSD detector was used. The mobile phase consisted of 0.1% HFBA and ACN at a flow rate of 1.0 mL min$^{-1}$. The following gradient was used for separation: initial conditions 100% B, 0% C; 0–1 min 95% B, 5% C; in 1–15 min 95–70% B, 5–30% C; in 15–20 min 70–60% B, 30–40% C (where "B" is 0.1% aqueous HFBA and "C" is ACN). The column was maintained at room temperature throughout the separation process, and the injected volume was 10 μL [34].

## Estimation of total soluble carbohydrate (TSC) and total soluble sugar (TSS) content

Freeze-dried leaf samples (25 mg) were homogenized in 5 mL of ethanol (95%). The homogenized samples were centrifuged at 5000 rpm for 10 min, and the supernatant was collected. The whole process was repeated with 70% ethanol, and the supernatant was kept in a refrigerator (4°C) for the analysis. TSC and TSS content was determined, according to the methods described by Khoyerdi et al. [35] and van Handel [36], respectively.

## Estimation of ascorbic acid (AsA) content

Ascorbic acid was isolated by extraction of the freeze-dried leaf sample (25 mg) with 6% trichloroacetic acid (10 mL), following the method described earlier [37]. The extract (4 mL) was mixed with 2 ml of dinitrophenyl hydrazine (2%), and one drop of 10% thiourea solution (in 70% ethanol). The mixture was boiled for 15 min in a water bath and allowed to cool to room temperature. After that, 5 mL of 80% $H_2SO_4$ (v/v) were added to the mixture at 0°C. The absorbance of the solution was read at 530 nm in a spectrophotometer and compared with a standard curve for AsA ranging from 10–100 mgL$^{-1}$.

## Determination of antioxidant enzyme activities and their gene expression

Leaves from all treated plants were collected and immersed immediately in liquid nitrogen and stored at -80°C until use. A 500 mg sample was homogenized in 5 mL of 50 mM sodium phosphate buffer solution (pH 7.8) using a pre-chilled mortar and pestle, then centrifuged at $15000 \times g$ for 20 min at 4°C. After collecting the supernatant, the enzyme extract was stored at 4°C for analysis. The activity of superoxide dismutase (SOD; EC 1.15.1.1) was estimated by the method described earlier [5]. The guaiacol peroxidase (GPX; EC 1.11.1.7) and catalase (CAT; EC 1.11.1.6) activities were determined according to Zhang [38]. The activity of ascorbate peroxidase (APX; EC 1.11.1.11) was assayed by the method developed earlier [39].

A quantitative reverse transcription-polymerase chain reaction (RT-qPCR) technique was used to determine the relative gene expression level of Cu/Zn-SOD, Fe-SOD, Mn-SOD, CAT, and APX using Actin as an internal control. Total RNA was extracted from leaves of sugar beet seedlings of all treatments, such as Drought, D + 0.3 mM Put, D + 0.6 mM Put, and D + 0.9 mM Put and control using an easy-spin$^{TM}$ kit (iNtRON Biotechnology, Korea). Briefly, approximately 100 mg leaf sample was powdered in liquid nitrogen. The sample was then homogenized with 1 ml lysis buffer and 200 μL of chloroform and centrifuged at $13000 \times g$ for 10 min. The lysates were mixed with washing buffer followed by elusion buffer to extract RNA through a mini spin column. cDNA was synthesized according to the QuantiTech reverse transcription procedure (Qiagen). In brief, the RNA extracts were rinsed with a Wipe buffer to remove genomic DNA and mixed with a master mixture containing reverse transcriptase, RT buffer, and RT primer mix. The mixture was incubated at 42°C for 15 min, followed by 95°C for 3 min. For qPCR assay, the PCR mixture (20 μl) containing 10 μl of 2 × QuantiTect SYBR Green PCR Master, 2 μl of each primer (250 μM), and 2 μl of cDNA, and 4 μl of RNase-free water was denatured at 95°C for 30 sec, followed by 45 cycles of 95°C for 5 sec, 55°C for 20 sec, and 72°C for 15 sec using a QuantStudio™ 3 Real-Time PCR System (Applied Biosystems™, USA). The oligonucleotide primers used in this study are listed in Table 1. The relative gene expression levels were estimated using a comparative method [40].

## Protein content

For calculation of enzyme activities, protein content was determined spectrophotometrically at 595 nm by the method of Bradford [41].

**Table 1. Primer sequences used for this study.**

| Gene | Molecular function | Primer sequence* | Melting temperature |
|------|--------------------|------------------|---------------------|
| Actin | Reference gene | F: 5′–TAAACCGAGATGGCTGATGC–3′ | 58.4 |
| | | R: 5′–ATACTTGGGAAGACAGCCCT–3′ | 58.4 |
| Cu/Zn-SOD | Redox dismutase and catalases | F: 5′–CTGTTTGCTTGCAGGTGGAC–3′ | 60.5 |
| | | R: 5′–AGACAAGCTTACCACAAGCC–3′ | 58.4 |
| Fe-SOD | Redox dismutase and catalases | F: 5′–AAGGGGCTTTGACTAGACCAT–3′ | 59.4 |
| | | R: 5′–ATGCTTTTGGCTTGCTGAGTG–3′ | 59.4 |
| Mn-SOD | Redox dismutase and catalases | F: 5′–GGGAGCATGCGTACTACCTT–3′ | 60.5 |
| | | R: 5′–AAACACAAGAATCTTCACCGGG–3′ | 60.3 |
| CAT | Redox dismutase and catalases | F: 5′–ATCATCCATGGAAGGCGTGA–3′ | 58.4 |
| | | R: 5′–TTGCTGGGTCCCATGATCG–3′ | 59.5 |
| APX | Redox ascorbate and glutathione ascorbate | F: 5′–GCAGCTTCTTTGGCACACAT–3′ | 58.4 |
| | | R: 5′–GAGCTTGAGAACCAGGCTTT–3′ | 58.4 |

*F, forward; R, reverse.

## Statistical analysis

All results were expressed as mean ± SD. The data were analyzed using SAS 9.4 (SAS Institute Inc., Cary, NC, USA) following a two-way analysis of variance, and the mean differences were compared by Tukey's post-hoc multiple comparison test. P values <0.05 were considered to be significant. The heatmap and clustering analysis were prepared by MetaboAnalyst 4.0 (www.metaboanalyst.ca), where samples were normalized by sum, and auto-scaling features were applied. Hierarchical cluster analysis was conducted using the Euclidean distance metric (average linking clustering). The principal component analysis (PCA) was carried out using OriginLab 10.0 (OriginLab, Northampton, MA, USA).

## Results and discussion

### Plant growth parameters

A significant (P $\leq$ 0.05) reduction in plant growth rate (PGR), shoot dry weight (SDW), and root dry weight (RDW) (39.44, 50, and 77.42%, respectively) was found in BSRI Sugar beet 2 in response to drought stress compared to the control (**Table 2**). However, the seedlings treated with 0.6 mM Put showed the best recovery in PGR (32.56%), SDW (41.86%), and RDW (171.4%). Although, the reduction in leaf area (LA), leaf relative water content (LRWC), and root-shoot ratio (RSR) were not significant under drought stress, LRWC increased significantly (37.78%) in the seedlings treated with 0.3 mM Put compared to the drought.

In the case of SBT-010, a significant (P $\leq$ 0.05) reduction was observed in PGR (62.5%) and LRWC (27.8%) under drought stress compared to control. However, the maximum recovery of PGR (214%), LA (40.20%) and LRWC (36.8%) were recorded by applying 0.9 mM Put, whereas maximum recovery of SDW (25.8%) was recorded by 0.6 mM Put. Significant improvement was observed only in PGR and LRWC.

Drought stress greatly inhibited the growth of seedlings; for instance, PGR, SDW, RDW, LA, and LRWC were significantly reduced when plants were grown under drought conditions (**Table 2**). A similar reduction in plant growth characteristics has previously been demonstrated in sugar beet [6], rice (*Oryza sativa* L.) [11], and *Thymus vulgaris* [28] under drought stress conditions. This growth retardation might be due to low

**Table 2. Effect of foliar application of putrescine (Put) on the plant growth rate (PGR), shoot dry weight (SDW), root dry weight (RDW), leaf area (LA), leaf relative water content (LRWC), and the root-shoot ratio (RSR) of sugar beet seedlings grown under drought stress conditions.**

| Genotypes | Treatments | Plant growth rate (cm/day) | Shoot dry weight (g) | Root dry weight (g) | Leaf area (cm$^2$) | LRWC (%) | Root-shoot ratio |
|---|---|---|---|---|---|---|---|
| | Control | 0.71 ± 0.07[a] | 0.86 ± 0.25[bc] | 0.31 ± 0.21[ab] | 61.76 ± 16.96 | 56.61 ± 1.71[bc] | 1.15 ± 0.13 |
| | Drought | 0.43 ± 0.11[bcd] | 0.43 ± 0.10[d] | 0.07 ± 0.03[c] | 39.70 ± 7.88 | 49.55 ± 3.86[c] | 1.21 ± 0.26 |
| BSRI Sugar beet 2 | D + 0.3 mM Put | 0.39 ± 0.15[cd] | 0.46 ± 0.11[d] | 0.12 ± 0.08[c] | 47.41 ± 9.18 | 68.27 ± 8.87[ab] | 1.25 ± 0.31 |
| | D + 0.6 mM Put | 0.57 ± 0.08[abc] | 0.61 ± 0.05[cd] | 0.19 ± 0.08[bc] | 53.44 ± 16.26 | 65.23 ± 2.98[ab] | 1.22 ± 0.31 |
| | D + 0.9 mM Put | 0.47 ± 0.12[abc] | 0.50 ± 0.10[d] | 0.18 ± 0.13[bc] | 44.67 ± 7.26 | 50.17 ± 0.91[c] | 1.30 ± 0.36 |
| | Control | 0.56 ± 0.17[abc] | 1.22 ± 0.20[a] | 0.38 ± 0.09[a] | 65.03 ± 13.82 | 69.70 ± 3.02[a] | 0.85 ± 0.14 |
| | Drought | 0.21 ± 0.09[d] | 0.93 ± 0.14[ab] | 0.23 ± 0.06[abc] | 46.55 ± 10.93 | 50.32 ± 1.06[c] | 1.04 ± 0.34 |
| SBT-010 | D + 0.3 mM Put | 0.52 ± 0.23[abc] | 0.96 ± 0.18[ab] | 0.19 ± 0.02[bc] | 56.04 ± 13.58 | 58.06 ± 2.44[abc] | 1.02 ± 0.30 |
| | D + 0.6 mM Put | 0.57 ± 0.11[abc] | 1.17 ± 0.18[ab] | 0.15 ± 0.09[bc] | 62.05 ± 14.31 | 56.56 ± 5.51[bc] | 1.12 ± 0.37 |
| | D + 0.9 mM Put | 0.66 ± 0.11[ab] | 0.95 ± 0.20[ab] | 0.20 ± 0.06[abc] | 65.26 ± 27.87 | 68.83 ± 9.48[a] | 0.95 ± 0.27 |
| HSD $_{(0.05)}$ | | 0.25 | 0.308 | 0.189 | NS | 12.12 | NS |

Values are mean ± SD of six replicates. Different letters show significant differences at P ≤0.05 (HSD).

photosynthetic activity, the osmotic imbalance caused cell dehydration, increased cell toxicity resulting from the accumulation of ROS, and lack of nutrient uptake [11, 42]. Drought also affects cell turgor and water uptake resulting in a lower accumulation of cell water, photo-assimilates, and metabolites related to cell elongation [28], resulting in a decrease in relative water content of the leaves [43]. Our study demonstrated that the exogenous application of 0.6 mM Put improved variables like PGR, SDW, RDW, LA, and LRWC in BSRI Sugar beet 2. In the case of SBT-010, the highest recovery of SDW and PGR resulted from spraying at 0.6 mM and 0.9 mM Put, respectively. The positive influence of exogenous Put on plant growth characteristics under drought stress has also been reported in *Thymus vulgaris* [28] and maize (*Zea mays* L.) [44]. Similarly, the positive impacts of exogenous application of polyamine, including Put against several abiotic stress, have been well documented in plants [23, 45, 46]. They may be due to the involvement of Put in several hormonal pathways, scavenging ROS, and the ability to adjust the osmotic balance [47–49]. It was reported that the effect of drought is more severe on aboveground parts than on roots, which often increases RSR in plants [50]. On the other hand, exogenous Put has been reported to improve the root biomass effectively in *Thymus vulgaris* [28]. In the present study, we observed a significant reduction of RDW in both sugar beet genotypes under drought, whereas no substantial recovery was observed by spraying Put on the plants. Furthermore, RSR did not show any significant response under drought stress or on Put application, in the present study.

## Photosynthetic pigments

Data presented in **Fig 1** showed that drought stress decreased photosynthetic pigments in both genotypes of sugar beet seedlings grown under water stress compared to the control condition. In BSRI Sugar beet 2, chlorophyll *a* (*Chl a*), chlorophyll *b* (*Chl b*), and carotenoid (*Car*) were significantly (P ≤ 0.05) reduced (10.8, 12.9, and 12.1%, respectively) under drought stress conditions. Exogenous Put at 0.3 mM restored *Chl a*, *Chl b*, and *Car* to 5.32, 8.65, and 3.67% in drought-stressed seedlings. Similar trends were observed in *Chl a* (14.8%) and *Chl b* (18.2%) for drought-stressed SBT-010, where maximum recovery (31.3 and 58.4%, respectively) was achieved by applying 0.3 mM Put. However, *Car* did not show any significant response to drought stress or Put treatment in SBT-010. Significant

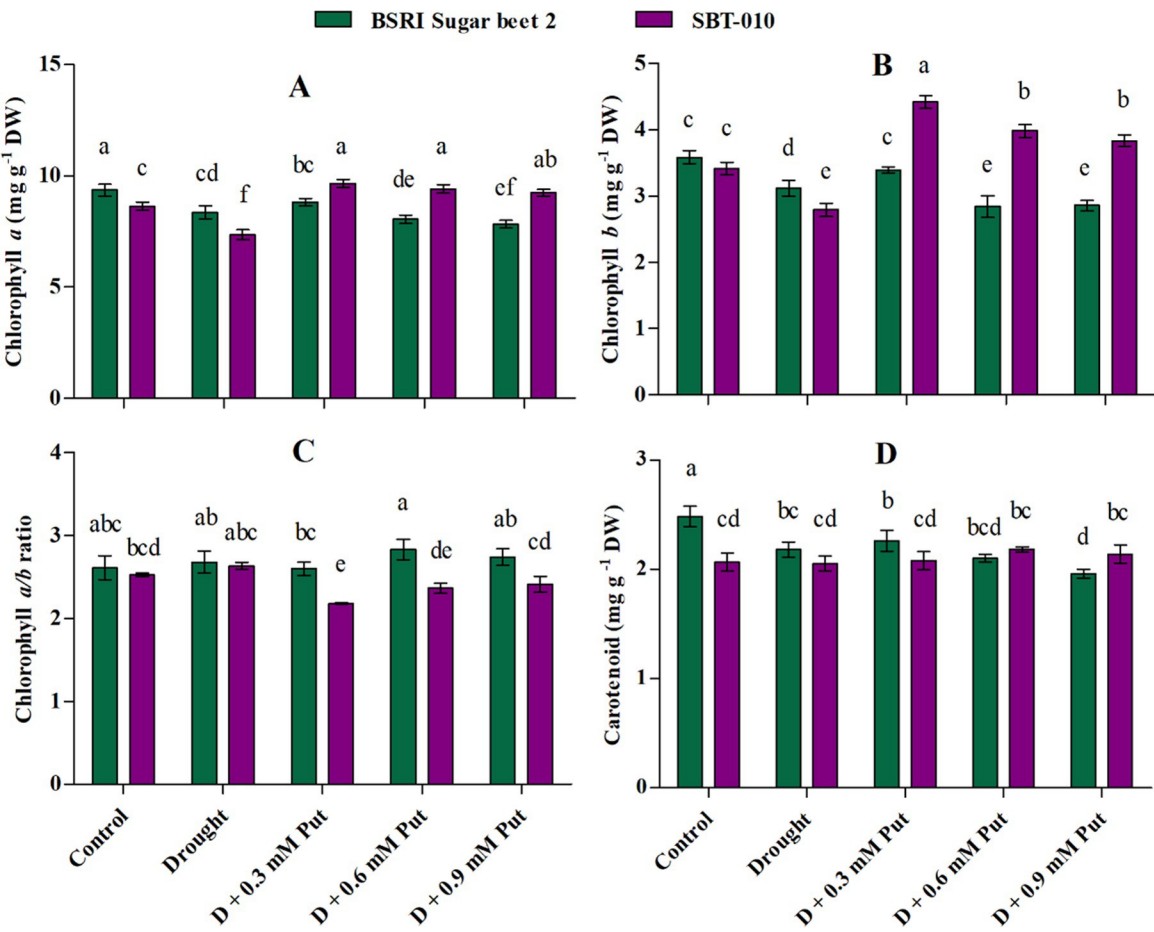

**Fig 1.** Effect of putrescine on photosynthetic pigments (A: *Chl a*, B: *Chl b*, C: *Chl a/b* ratio and D: Carotenoid) in leaves of two sugar beet genotypes (BSRI Sugar beet 2 and SBT-010) under drought stress condition. Data are expressed as the mean ± SD (n = 4). Statistical analysis was done using two-way ANOVA followed by Tukey's post-hoc multiple comparison test. Different letters indicate significant differences (P ≤ 0.05) among the treatments within each parameter.

responses were also absent for *Chl a/b* ratio in both genotypes, either by drought or Put treatments.

The pigment content showed a clear declining trend in *Chl a*, *Chl b*, and *Car* in sugar beet leaves (**Fig 1**). These symptoms were either due to rapid disruption or inhibition of the synthesis of photosynthetic pigments [51]. Drought enhances the synthesis of ROS, which leads to lipid peroxidation resulting in chlorophyll degradation [28, 52]. In the present experiment, spraying with Put at different concentrations improved *Chl a* & *b* concentrations in both genotypes, where the 0.3 mM level gave superior results. Similar results were also described in *Panax ginseng*, where 0.3 mM Put recovered the maximum photosynthetic pigments of saline-stressed seedlings [27]. Exogenous Put application enhances photosynthetic pigments, and these processes are well documented in various plant species [9, 45, 53, 54]. It is believed that Put protects thylakoid membranes through a chlorophyll-protein complex site and positively impacts chlorophyll levels in leaves [55]. A slight decreasing trend of the *Chl a/b* ratio under 0.3 mM Put could explain the increase of *Chl b*, which is well recognized for its protective role in the photosynthetic apparatus under drought stress.

**Table 3.** Effect of foliar application of putrescine (Put) on fluorescence intensity at 50 μs ($F_0$), maximal fluorescence intensity ($F_m$), maximum photosynthetic quantum yield (Fv/Fm), calculated PS II performance index (Pi_abs), and dissipated energy flux ($DI_0$/RC), of sugar beet seedlings grown under drought stress condition.

| Genotypes | Treatments | $F_0$ | $F_m$ | Fv/Fm | Pi_Abs | DIo/RC |
|---|---|---|---|---|---|---|
| | Control | 9098 ± 282.60[ab] | 39349 ± 2314 | 0.77 ± 0.01 | 2.07 ± 1.04 | 0.54 ± 0.05[ab] |
| | Drought | 8349 ± 1020.45[b] | 37057 ± 3455 | 0.76 ± 0.01 | 1.90 ± 0.59 | 0.52 ± 0.05[ab] |
| BSRI Sugar beet 2 | D + 0.3 mM Put | 8637 ± 1012.33[ab] | 38043 ± 4230 | 0.77 ± 0.04 | 2.04 ± 0.96 | 0.53 ± 0.14[ab] |
| | D + 0.6 mM Put | 8512 ± 1212.87[b] | 34424 ± 2454 | 0.75 ± 0.05 | 1.64 ± 1.04 | 0.58 ± 0.18[ab] |
| | D + 0.9 mM Put | 8610 ± 1091.85[ab] | 35198 ± 5424 | 0.75 ± 0.06 | 1.70 ± 1.47 | 0.57 ± 0.20[ab] |
| | Control | 10165 ± 693.02[a] | 39473 ± 3326 | 0.79 ± 0.02 | 2.05 ± 0.79 | 0.63 ± 0.12[a] |
| | Drought | 8257 ± 373.14[b] | 38845 ± 3058 | 0.74 ± 0.04 | 0.98 ± 0.52 | 0.45 ± 0.06[ab] |
| SBT-010 | D + 0.3 mM Put | 8128 ± 1173.83[b] | 38092 ± 3120 | 0.79 ± 0.03 | 2.22 ± 1.24 | 0.45 ± 0.12[ab] |
| | D + 0.6 mM Put | 7916 ± 389.43[b] | 37003 ± 3945 | 0.78 ± 0.03 | 1.96 ± 0.66 | 0.43 ± 0.06[ab] |
| | D + 0.9 mM Put | 7781 ± 257.41[b] | 37973 ± 3396 | 0.79 ± 0.02 | 2.16 ± 0.77 | 0.39 ± 0.05[b] |
| HSD (0.05) | | 1601.6 | NS | NS | NS | 0.22 |

Values are mean ± SD of six replicates. Different letters show significant differences at P ≤0.05 (HSD).

## Changes in fluorescence parameters

In the fluorescence induction kinetics (OJIP) parameters, fluorescence intensity at 50 μs ($F_0$), maximal fluorescence intensity ($F_m$), Fv/Fm, Pi_abs, and $DI_0$/RC were found to be low in both genotypes under drought conditions (**Table 3**), whereas a significant difference was observed for $F_0$ in SBT-010 only. In contrast, the application of Put at 0.3 mM concentration improved the Fv/Fm and Pi_abs considerably in both genotypes, but no significant change was observed for any variables in both genotypes.

Photosynthetic fluorescence is a byproduct of the photosynthetic process created by trapping light energy at the reaction center within the photosynthetic membrane, dissipating after photochemical activity with heat energy [56, 57]. It has also been reported that drought influences the photochemical activity of photosystem II (PS II) and electron requirement for photosynthesis. For this reason, an over-excitement occurs that results in photo-inhibition damage to the PS II reaction center [5]. In this study, drought substantially reduced all the OJIP parameters in both genotypes, while applying 0.3 mM Put improved the Fv/Fm and Pi_abs considerably. The maximum recovery of $F_0$ and Fm were also recorded in BSRI Sugar beet 2 by applying 0.3 mM Put. In a previous study, $F_0$, Fm, and Fv/Fm were reported to be significantly increased by exogenous Put application in the Iranian mandarin Bakraii (*Citrus reticulata× Citrus limetta*) under salinity stress [58]. In some other studies, higher Fv/Fm was denoted as a stress tolerance indicator under cold stress [59, 60], salinity stress [17, 61], and drought stress [7, 62]. Plants with higher Pi_abs and $DI_0$/RC have also demonstrated higher tolerance to drought stress [63]. It is believed that under drought conditions, photosynthetic pigments of photosystems are damaged by stress factors resulting in a low light-absorbing efficiency in PS I and PS II. This reduced light-absorbing efficiency is the prime cause of reduced photosynthetic capacity in plants [64]. Our present study also supports this finding as drought stress reduced the photosynthetic pigments and photosynthetic capacity (photosynthetic rate and photosynthetic quantum yield) in both genotypes.

## Photosynthetic parameters

A significant reduction (P ≤ 0.05) in photosynthetic rate (Pn), transpiration rate (E), and stomatal conductance ($g_s$) were recorded in BSRI Sugar beet 2 under drought conditions (**Fig 2**). Drought expressively reduced Pn, E, and $g_s$ at 66.29, 78.66, and 88.88%, respectively. In

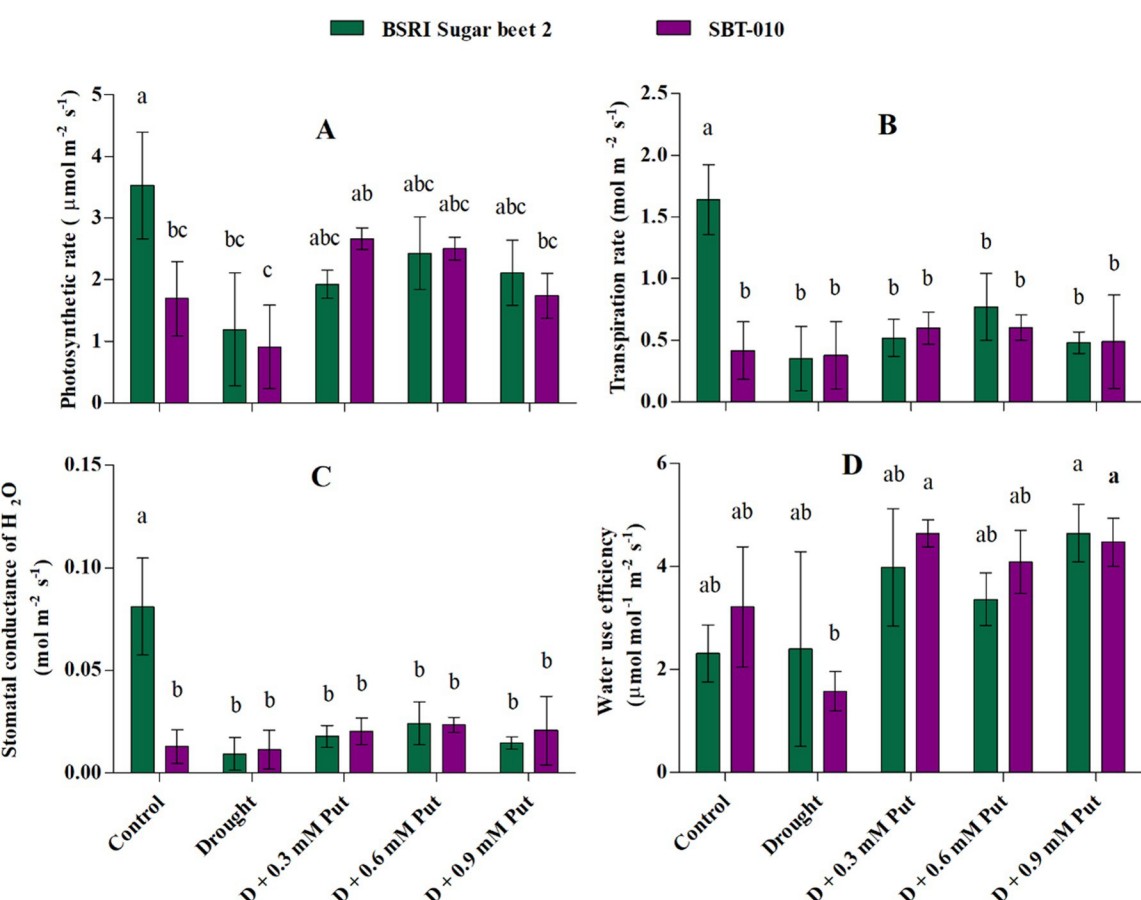

**Fig 2.** Effect of putrescine on photosynthetic gas exchange (A: photosynthetic rate, B: transpiration, C: stomatal conductance, and D: water use efficiency) in leaves of two sugar beet genotypes (BSRI Sugar beet 2 and SBT-010) under drought stress condition. Each value represents the mean ± SD (n = 6). Statistical analysis was done using two-way ANOVA followed by Tukey's post-hoc multiple comparison test. Different letters indicate significant differences (P ≤ 0.05) among the treatments within each parameter.

contrast, no significant changes were observed in any photosynthetic parameters under drought stress in SBT-010. Although drought-stressed seedlings treated with exogenous Put displayed an increasing trend in both genotypes for all parameters, significant responses were recorded in SBT-010 only for Pn and water use efficiency (WUE) on the application of 0.3 mM Put.

Drought restricts metabolic activities and causes low $CO_2$ diffusion to the chloroplast resulting in lower photosynthesis [65]. Generally, mild to moderate water stress creates stomatal limitation, which is considered an important reason for lower photosynthetic activities under such conditions [65, 66]. Photosynthetic rate, transpiration, stomatal conductance, and water use efficiency decreased in response to drought in both sugar beet cultivars. Similar results were also observed in some previous studies [65, 67], where the photosynthetic rate and stomatal conductance were significantly reduced by drought stress. The decreased photosynthetic rate may be due to the higher accumulation of ROS (MDA and $H_2O_2$), as they maintained a negative correlation with each other (Fig 6). In contrast, the application of 0.3 mM Put improved photosynthetic capacity at 61.3% and 191% with a substantial decrease of ROS in drought-stressed BSRI Sugar beet 2 and SBT-010, respectively. Similar results were also reported in a previous study where Put significantly improved the photosynthesis and WUE with substantially declined $H_2O_2$ and

MDA in drought-affected rice seedlings [68]. The results indicated that the higher photosynthetic capacity was associated with less oxidative injury, where Put improved photosynthesis capacity in drought-stressed sugar beet seedlings by reducing ROS and cell injury.

## Malondialdehyde and $H_2O_2$ concentration

The effect of drought on lipid peroxidation or cellular damage indicator (MDA) is shown in **Fig 3A**. MDA levels were raised significantly ($P \leq 0.05$) in BSRI Sugar beet 2 and SBT-010 (27.1 and 35.6%, respectively) under drought stress conditions. However, exogenous Put application significantly reduced MDA levels in both genotypes. The lowest MDA values were recorded in the 0.3 mM, and 0.6 mM Put treated BSRI sugar beet 2 (2.76 μmol g$^{-1}$ FW) and SBT-010 (2.90 μmol g$^{-1}$ FW), respectively, under drought stress conditions.

Drought also significantly changed the concentration of $H_2O_2$ ($P \leq 0.05$). $H_2O_2$ levels were increased by 118% and 89.4% in drought-stressed BSRI Sugar beet 2 and SBT-010, respectively (**Fig 3B**). Like MDA, the level of $H_2O_2$ decreased with Put treatments, particularly the lowest

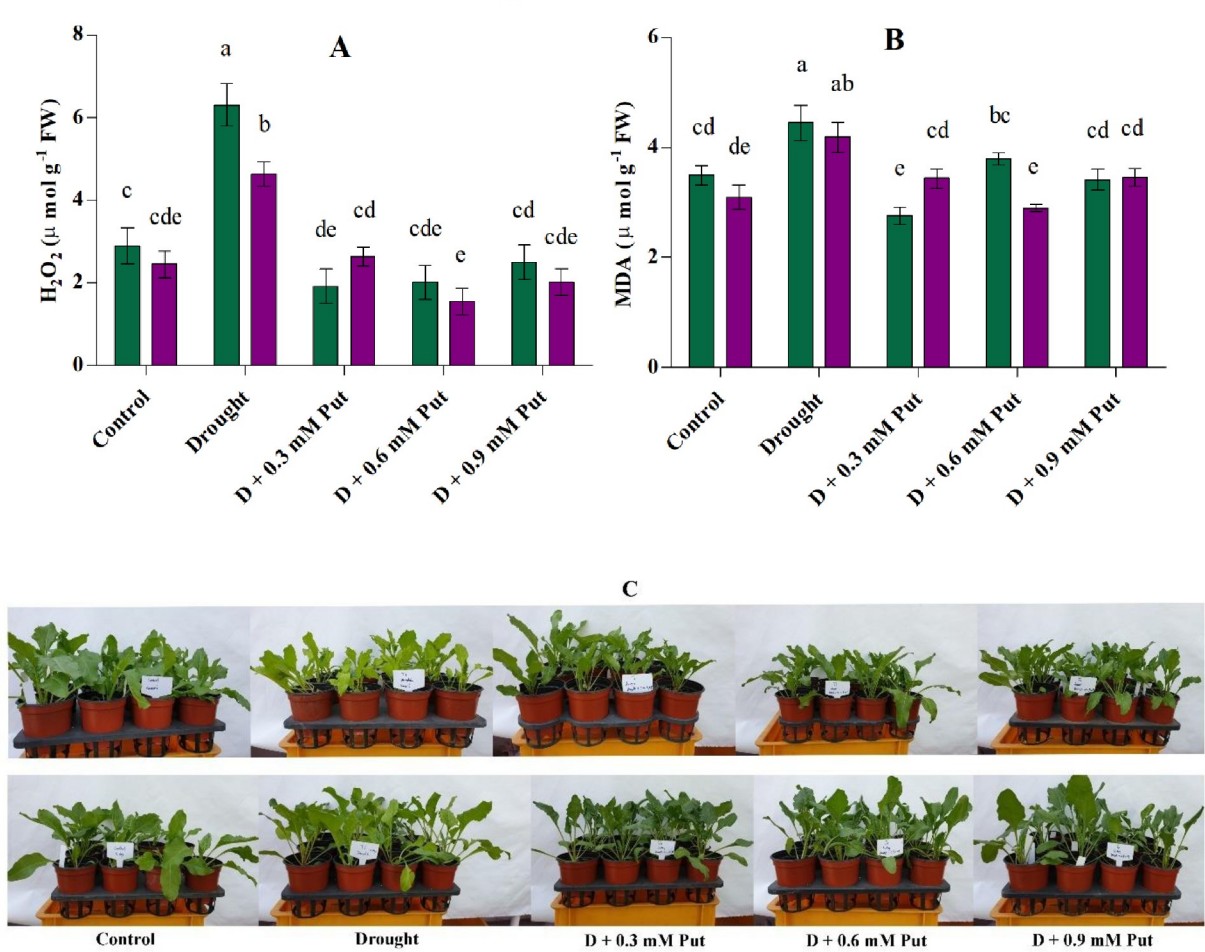

**Fig 3.** Effect of putrescine on $H_2O_2$ (A, hydrogen peroxide), MDA (B, malondialdehyde) content in leaves and phenological appearance (C) of two sugar beet genotypes (BSRI Sugar beet 2 and SBT-010) grown under drought stress condition. Each value represents the mean ± SD (n = 4). Statistical analysis was done using two-way ANOVA followed by Tukey's post-hoc multiple comparison test. Different letters indicate significant differences ($P \leq 0.05$) among the treatments within each parameter.

values recorded in the 0.3 mM and 0.6 mM Put treated BSRI sugar beet 2 (1.92 µmol g$^{-1}$ FW) and SBT-010 (1.54 µmol g$^{-1}$ FW) respectively.

Drought is well recognized for provoking ROS generation. Excess ROS creates oxidative stress in plant cellswhere lipids and proteins are the primary victims [69]. Oxidative stress by drought is mainly induced by interrupting electron flow in the photosynthesis process, which is predominantly responsible for ROS generation [70]. In the present study, drought stress substantially increased $H_2O_2$ and MDA in both genotypes. The elevation of ROS level in response to drought in sugar beet was also reported in previous studies [5, 6, 67]. Generally, MDA levels, commonly used to determine the quantity of lipid peroxidation, were responsible for photosynthetic pigments disorganization, protein denaturation, enzyme activity inhibition, and finally programmed cell death [69, 71–73]. However, depending on the concentration and ROS scavenging mechanism, $H_2O_2$ can act as both a cell-damaging and signaling molecule [5, 74, 75]. Furthermore, MDA and $H_2O_2$ are responsible for lowering plant growth and development [11]. In the present study, exogenous application of 0.3 mM Put at significantly reduced MDA and $H_2O_2$ to 38.0% and 69.9% in BSRI Sugar beet 2, and 17.9% and 43.2% in SBT-010 compared to the seedlings treated with drought stress only. The better phenological appearance (**Fig 3C**) of the Put treated drought-stressed seedlings is correlated with the MDA and $H_2O_2$ levels. The reduced MDA and $H_2O_2$ accumulation may result from a higher concentration of osmolytes and activity of antioxidant enzymes caused by exogenous Put in drought-stressed sugar beet seedlings (**Table 4 and Fig 4**). Several earlier studies have claimed that exogenous Put application modulates plant growth and the photosynthetic apparatus, and stimulates antioxidant capacity and gene expression under different abiotic stresses [21, 28, 47, 53, 76].

## Osmotic adjustment molecules

The results indicated that proline (Pro), glycine betaine (GB), mannitol, and ascorbic acid (AsA) were affected significantly by drought stress (**Table 4**). Drought stress significantly ($P \leq 0.05$) increased the content of Pro and GB in both genotypes compared to the control. On the other hand, a significant ($P \leq 0.05$) reduction was observed in mannitol and AsA for SBT-010. Results also indicated that Put simultaneously increased Pro with the foliar application at 0.6 mM concentration while TSS and AsA at 0.3 mM concentration in drought-affected seedlings of both genotypes. In contrast, GB and mannitol were reduced by 0.3 mM Put on

**Table 4. Effects of exogenous putrescine on osmolytes of two sugar beet genotypes under drought stress conditions.**

| Genotypes | Treatments | Proline (µmole g$^{-1}$ DW) | Glycine betaine (mg g$^{-1}$ FW) | Mannitol (mg g$^{-1}$ FW) | TSC (mg g$^{-1}$DW) | TSS (mg g$^{-1}$ DW) | AsA (mg g$^{-1}$ FW) |
|---|---|---|---|---|---|---|---|
| BSRI Sugar beet 2 | Control | 13.19 ± 0.03[h] | 1.77 ± 0.03[e] | 4.98 ± 0.06[b] | 0.53 ± 0.05 | 0.12 ± 0.02[b] | 1.81 ± 0.04[ef] |
| | Drought | 21.77 ± 0.13[d] | 3.40 ± 0.11[a] | 4.75 ± 0.02[bc] | 0.47 ± 0.06 | 0.13 ± 0.02[b] | 1.74 ± 0.03[f] |
| | D + 0.3 mM Put | 22.97 ± 0.16[c] | 2.88 ± 0.08[b] | 4.47 ± 0.04[c] | 0.47 ± 0.08 | 0.19 ± 0.03[a] | 2.00 ± 0.03[b] |
| | D + 0.6 mM Put | 27.33 ± 0.12[a] | 2.58 ± 0.08[cd] | 4.43 ± 0.04[c] | 0.51 ± 0.10 | 0.15 ± 0.002[ab] | 1.95 ± 0.02[bc] |
| | D + 0.9 mM Put | 24.78 ± 0.17[b] | 2.38 ± 0.08[d] | 4.01 ± 0.04[d] | 0.50 ± 0.10 | 0.14 ± 0.02[b] | 2.09 ± 0.05[a] |
| SBT-010 | Control | 4.81 ± 0.48[i] | 1.93 ± 0.06[e] | 7.23 ± 0.21[a] | 0.54 ± 0.07 | 0.12 ± 0.001[b] | 1.84 ± 0.03[de] |
| | Drought | 13.86 ± 0.11[g] | 3.26 ± 0.09[a] | 3.54 ± 0.11[e] | 0.49 ± 0.09 | 0.15 ± 0.003[b] | 1.58 ± 0.01[g] |
| | D + 0.3 mM Put | 15.05 ± 0.07[f] | 2.95 ± 0.09[b] | 3.38 ± 0.22[g] | 0.40 ± 0.07 | 0.16 ± 0.002[ab] | 1.91 ± 0.01[cd] |
| | D + 0.6 mM Put | 16.49 ± 0.03[e] | 2.91 ± 0.09[b] | 2.99 ± 0.15[f] | 0.53 ± 0.05 | 0.16 ± 0.007[ab] | 1.90 ± 0.02[cd] |
| | D + 0.9 mM Put | 16.83 ± 0.09[e] | 2.78 ± 0.07[bc] | 2.27 ± 0.12[e] | 0.39 ± 0.02 | 0.13 ± 0.019[b] | 1.97 ± 0.002[bc] |
| HSD $_{(0.05)}$ | | 0.44 | 0.23 | 0.36 | NS | 0.04 | 0.08 |

Values are mean ± SD of three replicates. Different letters show significant differences at P ≤0.05 (HSD).

drought-stressed seedlings. Besides, neither drought nor exogenous Put caused any significant change in TSC accumulation in both genotypes.

Drought stress affects ROS homeostasis by stimulating overproduction, resulting in oxidative damage to plants. To protect from such damage, the plant accumulates Pro under stressful conditions [77]. It was also found that Pro can detoxify ROS, especially $^{\bullet}$OH, enhance photochemical activity in thylakoid membranes, and reduce MDA formation under several abiotic stresses [78–81]. A higher accumulation of Pro concentration is an indicator of the plants' adaptive drought response [82]. GB is an amphoteric compound playing a vital role in the osmotic adjustment of sugar beet plants under stress conditions [83]. GB stabilizes protein structure, regulates the enzymatic activity and gene transcription, and acts as an osmolyte to maintain cellular volume to protect cells under stress [84]. However, mannitol, TSC, TSS, and AsA have previously been described as osmoprotectants [9, 85, 86]. Pro, GB, and TSS accumulation increased 92.9%, 92.1%, and 8.3% in BSRI sugar beet 2 whereas 51.8%, 68.9%, and 25% in SBT-010 genotypes, respectively, under drought stress compared to the control (**Table 4**).

In contrast, a foliar application of Put increased the concentration of Pro and AsA while decreasing GB significantly compared to drought in both genotypes. Aditionally, a significant reduction of mannitol in SBT-010 and enhancement of TSS in BSRI Sugar beet 2 were also recorded under Put treatment. Similar findings were reported in wheat under drought stress, where exogenous Put positively influenced Pro, soluble, and insoluble sugar accumulation [87]. In the present experiment, a maximum Pro and TSS accumulation were observed by exogenous application of Put at 0.6 mM and 0.3 mM concentrations, respectively, in both genotypes compared to drought conditions. These findings suggested that exogenous Put might reduce osmotic stress by causing a change in the concentrations of different osmolytes.

Mannitol (a six-carbon acyclic polyol) is considered a crucial osmoprotectant that has a key role in the photosynthesis process and abiotic stress tolerance [88]. In the present study, we did not observe any increment in mannitol concentration by exogenous Put treatment. Both mannitol and AsA declined under drought stress, where AsA was significantly improved in both genotypes by 0.3 mM Put treatment. There is evidence that mannitol can be accumulated under different abiotic stresses in various plant species other than halophytes [88]. Therefore, being a halophytic crop [89], sugar beet showed a negative trend in mannitol accumulation in the present study under all treatments. Notably, AsA was significantly increased 14.9 and 20.9% by 0.3 mM Put application in BSRI Sugar beet 2 and SBT-010, respectively, compared to drought stress. Although AsA plays a vital role in protecting plants from osmotic stress, its elevation under stress conditions is particularly species-specific [90]. For instance, decreasing trends of AsA were reported under drought stress in spinach but not in soybean leaves [90, 91].

## Antioxidant enzyme activity

The effects of drought and application of Put at different concentrations are shown in **Fig 4**. Compared with the control, the activities of SOD were significantly ($P \leq 0.05$) reduced in BSRI Sugar beet 2 at 16.99% under drought stress. In the case of BST-010, the activities of SOD, CAT, and GPX were significantly ($P \leq 0.05$) reduced (37.8, 80.8, and 40.7%, respectively) under drought stress. Furthermore, the application of 0.3 mM Put apparently improved the activities of SOD (30.0%) and CAT (28.0%) in BSRI Sugar beet 2. In the case of SBT-010, the activities of SOD and GPX substantially improved (13.2 and 10.0%, respectively) by 0.6 mM Put, while CAT activity improved most (191.9%) by 0.3 mM Put application.

The severity of oxidative damage due to excess ROS generation can be modulated by the up-regulation of enzymatic and non-enzymatic antioxidants such as SOD, CAT, APX, GPX, AsA, carotenoids, etc. [5, 11]. The most common ROS produced in response to stress

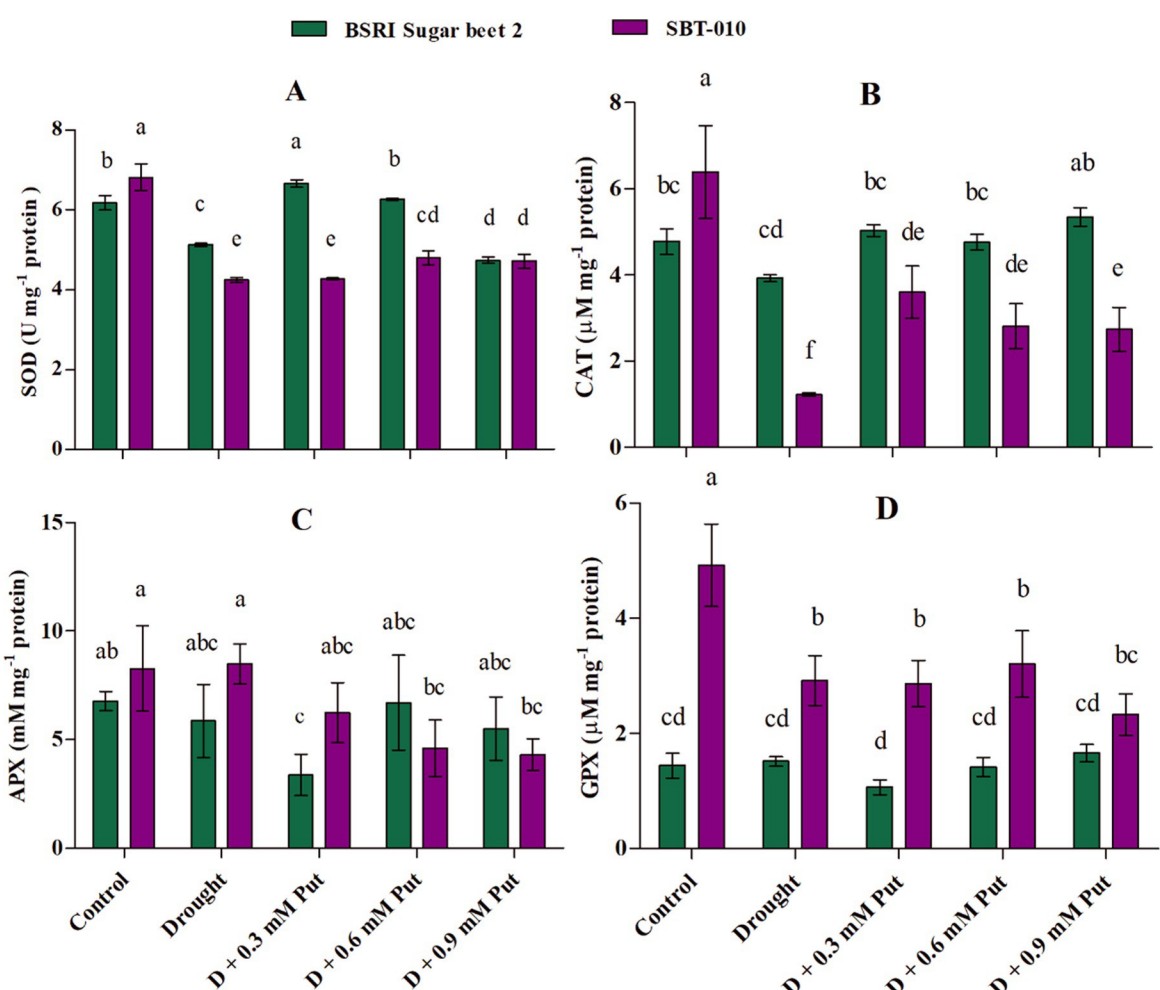

**Fig 4.** Effect of putrescine on antioxidant enzymes (A: Superoxide dismutase, SOD; B: Catalase, CAT; C: Ascorbate peroxidase, APX; and D: Guaiacol peroxidase, GPX) activities in leaves of two sugar beet genotypes (BSRI Sugar beet 2 and SBT-010) under drought stress condition. Each value represents the mean ± SD (n = 4). Statistical analysis was done using two-way ANOVA followed by Tukey's post-hoc multiple comparison test. Different letters indicate significant differences (P ≤ 0.05) among the treatments within each parameter.

conditions is $H_2O_2$ [92], which is considered to be the result of the reaction of the $O_2^{-\bullet}$ catalyzed by cellular SOD [93, 94]. In general, drought stress causes oxidative damage due to decreasing antioxidant enzyme activities and increased lipid peroxidation [95]. In the present study, the activity of SOD and CAT decreased in BSRI Sugar beet 2, whereas SOD, CAT, and GPX were reduced in SBT-010 under drought stress, and upregulated by exogenous application of Put.

However, it seems that reduced antioxidant enzymes activities in drought stress seedlings are mainly responsible for excess ROS, as we observed higher $H_2O_2$ and MDA accumulation under drought stress (**Fig 3**), which is similar to previous findings [5, 11, 27]. Generally, the plants exposed to stress triggered the ROS due to an imbalance of ROS accumulation and the scavenging system resulting in cellular oxidative damage in plants [27]. In the present study, SOD, CAT, and GPX were found most vulnerable in the leaves of SBT-010 under drought stress compared to the BSRI Sugar beet 2. Generally, drought induces highly intensive oxidative stress resulting in loss of cell turgor, which may cause the failure of the plant defense mechanisms [67]. In contrast, osmolytes like Pro, TSC, TSS, and AsA accumulation were

increased by applying Put to drought-stressed plants (**Table 4**). In this connection, a previous study has also described the correlation of enzymatic activity with Pro accumulation [77]. The decreased ROS with higher accumulation of osmolytes and higher enzymatic activities by exogenous Put application elucidated a more balanced condition in drought-stressed seedlings that improved the morpho-physiological parameters. Some earlier studies have claimed that exogenous Put application stimulates antioxidant enzyme activity and stress response gene expression of plants under several different abiotic stresses [28, 53, 96].

## Relative gene expression

The relative expression level of Cu/Zn-SOD, CAT, and APX genes increased while Fe-SOD and Mn-SOD genes decreased under drought stress in BSRI Sugar beet 2 compared to the control (**Fig 5**). Putrescine application upregulated the expression of all genes except CAT in the seedlings under drought stress. The expression level of Cu/Zn-SOD, Fe-SOD, and APX genes was observed to be highest at the 0.6 mM level and Mn-SOD at the 0.3 mM level. In the case of SBT-010, Cu/Zn-SOD and CAT transcript levels were decreased, while Fe-SOD, Mn-SOD, and APX increased under drought stress. Importantly, Put application upregulated the transcript level of Mn-SOD, CAT, and APX under drought stress conditions. The best results for CAT and APX were observed at 0.3 mM concentration and for Mn-SOD at 0.9 mM concentration, respectively.

The mechanisms related to the antioxidant defense system, which increased the drought tolerance of the plant, may reflect their gene transcription level [14]. The higher expression of SOD, CAT, and APX genes has been reported to improve oxidative stress tolerance in plants [97, 98]. In the present study, the levels of expression of Cu/Zn-SOD, Fe-SOD, and APX genes were increased most, by 4.8, 3.0, and 3.83 fold compared to the control, in the BSRI Sugar beet 2 when treated with 0.6 mM concentration of Put. CAT gene expression increased in all treatments compared to the control, where Put did not make any significant improvement in drought-stressed seedlings. Mn-SOD genes were down-regulated in all cases except in plants treated with 0.3 mM Put (2.83 fold). On the other hand, in the case of SBT-010, the Mn-SOD gene was upregulated in all cases compared to the control. CAT and APX genes were increased to the highest level by the application of 0.3 mM Put concentration, whereas Cu/Zn-SOD and Fe-SOD genes were not upregulated by Put treatment. The antioxidant defense of plants is known to affect the level of stress responses induced by drought and salinity. The role of APX in this process is also evidenced by transcriptome studies [14]. Additionally, the over-expression of Cu/Zn-SOD and CAT have been reported to strengthen the antioxidant system and increased the tolerance level of plants under drought conditions [99, 100]. The variation in the expression level of Cu/Zn-SOD, Fe-SOD, Mn-SOD, and APX genes was also reported in sugar beet seedlings under salinity stress conditions [17, 101]. In the present study, 0.3 mM Put concentration manifested higher expression of the Cu/Zn-SOD, Fe-SOD, Mn-SOD, and APX genes when both genotypes were considered. The expression pattern of the APX gene did not show similarities with the enzymatic activity. In this connection, previous reports depicted a discrepancy in the relationship between the transcript levels and related enzymes activity in some cases [102–104]. This imparity during drought stress explained that enzyme changes regulate at the post-transcriptional level and not at mRNA levels [14]. Overall, Put treatment at 0.3 mM concentration was found to modulate the gene expression of all SOD, CAT, and APX enzymes in both genotypes.

## Hierarchical clustering and PCA analysis

The morpho-physiological and biochemical data from both sugar beet genotypes under all treatments were employed to construct a heatmap, hierarchical clustering, and PCA. In both

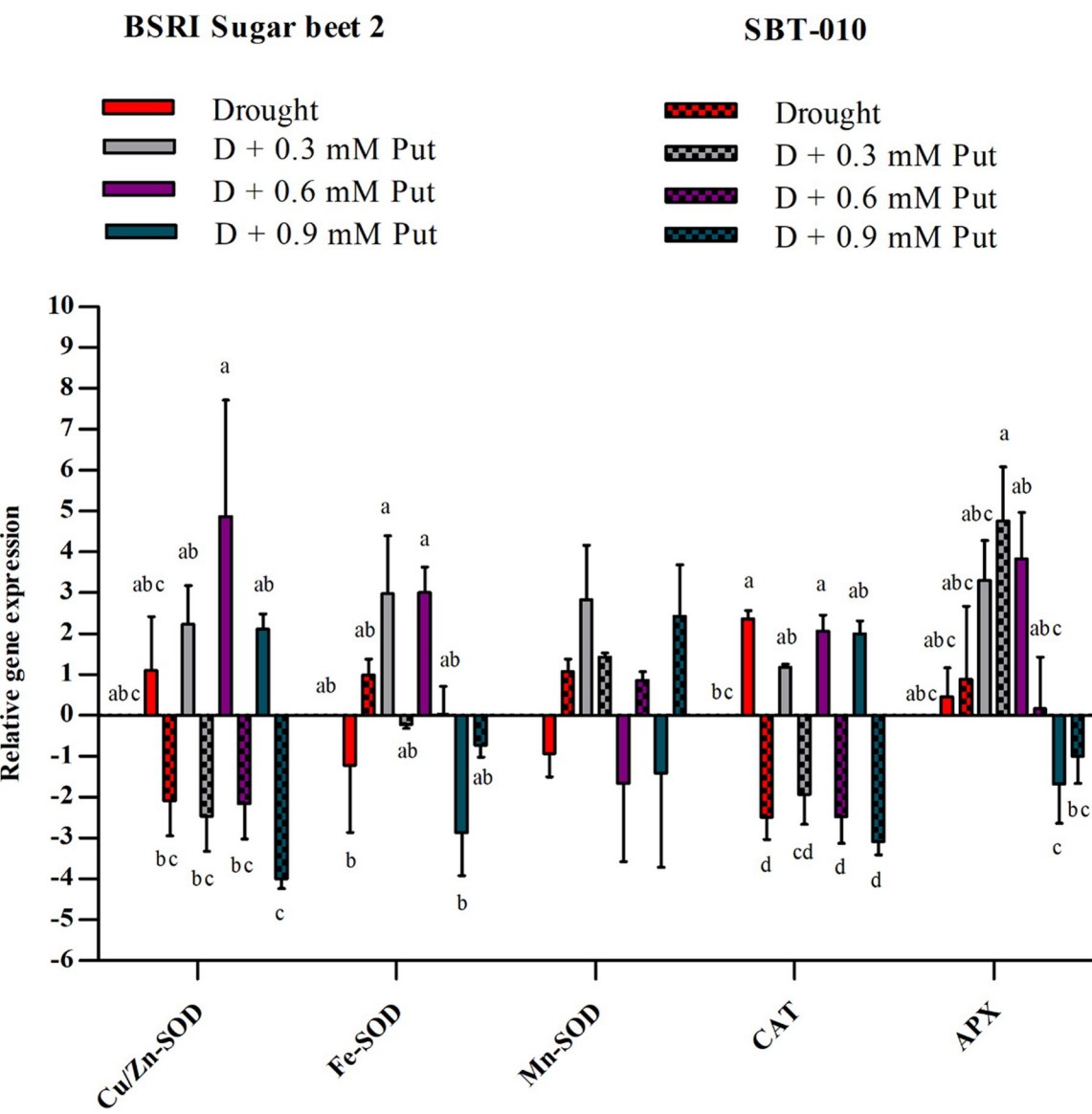

**Fig 5. The relative expression of the five genes (Cu/Zn-SOD; Fe-SOD; Mn-SOD; CAT and APX) compare to control in leaves of two sugar beet genotypes (BSRI Sugar beet 2 and SBT-010) under Drought; Drought + 0.3 mM putrescine; Drought + 0.6 mM putrescine; Drought + 0.9 mM putrescine.** Each value represents the mean ± SD (n = 3). Statistical analysis was done using two-way ANOVA followed by Tukey's post-hoc multiple comparison test. Different letters indicate significant differences (P ≤ 0.05) among the treatments within each parameter.

genotypes, hierarchical clustering grouped all the variables into two clusters (cluster-A and cluster-B) (Fig 6A). In the case of BSRI Sugar beet 2, some biochemical and stress response parameters like Pro, GB, AsA, TSS, MDA, $H_2O_2$, SOD, CAT, APX, and GPX with some morphological traits like LRWC, *Ch la/b* ratio, root-shoot ratio, and WUE were clustered into cluster-A. In the heatmap, the parameters SOD, CAT, AsA, and LRWC displayed a decreasing trend, and the parameters $H_2O_2$, MDA, Pro, GB, *Chl a/b* ratio, root-shoot ratio, and GPX displayed an increasing trend in drought-stressed seedlings compared to the control. However, the parameters $H_2O_2$ and MDA were downregulated, and SOD, TSS, and LRWC were upregulated most by the application of exogenous Put at 0.3 mM concentration. Furthermore, an

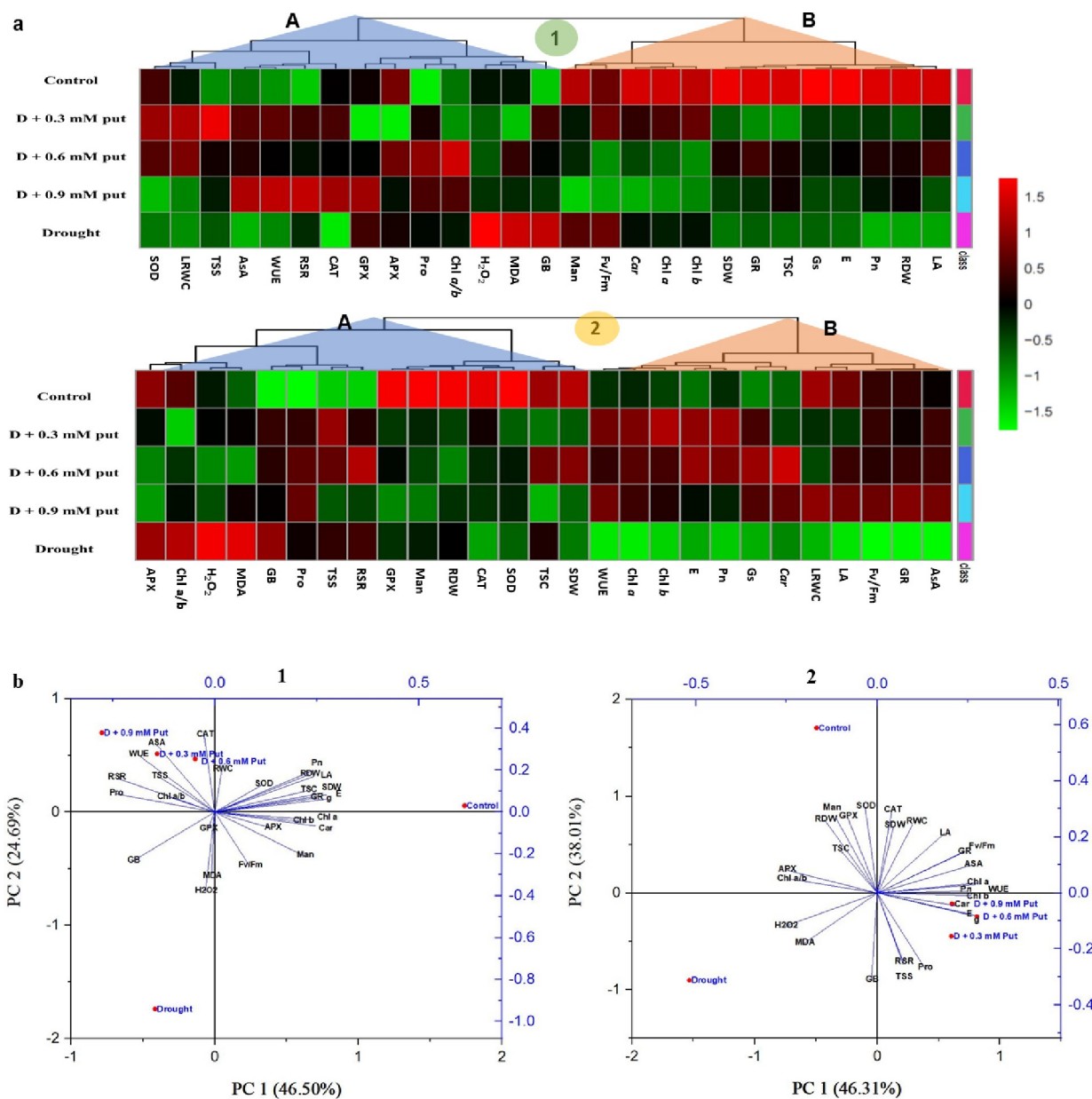

**Fig 6.** Hierarchical clustering and heatmap analysis (a), and principal component analysis (PCA) (b) to elucidate the variable treatment relationships under five treatments for 10 days. In the heatmap, the mean values of the various parameters obtained in this study were normalized and clustered. At the variable level, two separate clusters were recognized for each genotype (1. BSRI Sugar beet 2, and 2. SBT-010). The color scale displays the intensity of normalized mean values of different parameters. In PCA, the lines starting from the central point of the biplots display negative or positive associations of different variables, and their proximity specifies the degree of correlation with specific treatment (1. BSRI Sugar beet 2, and 2. SBT-010). Control (Field capacity); Drought (D, 30% moisture level); D + 0.3 mM putrescine; D + 0.6 mM putrescine; D + 0.9 mM putrescine. GR, Growth rate; SDW, Plant dry weight; RDW, Root dry weight; LA, Leaf area; LRWC, Leaf Relative water content; RSR, Root shoot ratio; *Chl a*, chlorophyll *a*; *Chl b*, chlorophyll *b*; *Car*, carotenoid; Chl *a/b*, *Chl a/b* ratio; Fv/Fm, photosynthetic quantum yield; Pn, photosynthetic rate; E, transpiration rate; Gs, stomatal conductance; WUE, water use efficiency; $H_2O_2$, hydrogen peroxide content; MDA, malondialdehyde content; Pro, proline content; GB, glycine betaine content; Man, mannitol content; TSC, total soluble carbohydrate; TSS, total soluble sugar content; AsA, ascorbic acid; SOD, superoxide dismutase; CAT, catalase activity; APX, ascorbate peroxidase; GPX, guaiacol peroxidase.

upregulation trend of Pro, APX, and *Chl a/b* ratio by 0.6 mM concentration and AsA, WUE, root-shoot ratio, CAT, and GPX by 0.9 mM concentration of Put were also displayed in cluster-A. On the other hand, cluster-B represents mostly morphological (GR, LA, SDW, and

RDW) and photosynthetic attributes (Pn, E, Gs, Fv/Fm, *Chl a*, *Chl b*, and *Car*) and some biochemical parameters, including TSC and mannitol. All the cluster-B variables were downregulated by drought stress compared to the control, and they showed a rising trend when treated with exogenous Put. A 0.3 mM concentration of Put upregulated the parameters like *Chl a*, *Chl b*, *Car*, and Fv/Fm, while 0.6 mM concentration upregulated the parameters LA, GR, SDW, RDW, Pn, E, Gs, and TSC. In the case of SBT-010, some biochemical and stress response parameters like Pro, GB, TSC, TSS, mannitol, MDA, $H_2O_2$, SOD, CAT, APX, and GPX with some morphological traits (SDW, RDW, *Chl a/b* ratio, and root-shoot ratio) were clustered into cluster-B. In the heatmap, the parameters $H_2O_2$, MDA, Pro, GB, TSS, APX, *Chl a/b* ratio, and root-shoot ratio displayed an increasing trend while the parameters SOD, CAT, GPX, SDW, RDW, TSC, and mannitol displayed a decreasing trend in drought-stressed seedlings compared to the control. However, the parameters $H_2O_2$ and MDA were downregulated, and SOD, GPX, Pro, TSC, and SDW were upregulated by the application of exogenous Put at 0.6 mM concentration. In contrast, Cluster-B represents mostly photosynthetic attributes (Pn, E, Gs, WUE, Fv/Fm, *Chl a*, *Chl b*, and *Car*), morphological (GR, LA, LRWC), and osmolyte TSC. All the variables in cluster-B were significantly downregulated by drought stress compared to the control, and they showed a rising trend when treated with different Put concentrations. However, parameters like *Chl a*, *Chl b*, Pn, E, and WUE were upregulated most by the application of exogenous Put at 0.3 mM concentration, while Gs and *Car* were upregulated most by 0.6 mM and LRWC, LA, Fv/Fm, GR, and AsA were upregulated by 0.9 mM Put.

PCA analysis was carried out to uncover the connections between the different parameters in different treatment groups (**Fig 6B**). The elements of PC1 and PC2 together described 71.2% and 84.3% of the variability in BSRI Sugar beet 2 and SBT-010, respectively. The treatment control and drought manifested an opposite relationship in both genotypes, where control had a close association with most morphological and biochemical parameters, and drought had an intimate association with MDA and $H_2O_2$. Interestingly, exogenous Put in drought-stressed plants showed a close relationship with the control in both genotypes. The PCA also indicated that drought-stressed sugar beet seedlings showed a positive relationship with ROS ($H_2O_2$ and MDA). In contrast, Put-treated drought-stressed seedlings showed a positive relationship with growth-related parameters, suggesting protective roles of Put in diminishing the toxic results of drought on sugar beet seedling growth and development.

The schematic diagram (**Fig 7**) represents that drought exerts its harmful effects on plants by reducing the *Chl* content resulting in decreased photosynthetic rates. Drought stress increases $O_2^{-\bullet}$, which in turn increases $H_2O_2$ content. Water scarcity induces lower levels of AsA, mannitol, and *Car* with reduced activity of SOD, CAT, and GPX enzymes resulting in boosting of $H_2O_2$. Drought also enhances MDA, which damages the cell membrane and promotes lipid peroxidation. LA and LRWC also decrease in seedlings under drought, limiting gas exchange and photosynthesis. On the other hand, exogenous Put treatment can restore the growth of drought-stressed seedlings and reduce oxidative damage. Exogenous Put induced increase in Pro, TSC, TSS, AsA, and *Car* content along with antioxidant enzymes like SOD, CAT, and GPX maintain the optimum level of ROS. Put reduces the cell membrane damage by lowering lipid peroxidation, which increases *Chl* content, LA, and LRWC in seedlings, thus maintaining higher photosynthesis and biomass accumulation.

## Conclusions

Drought stress caused oxidative stress in sugar beet seedlings due to the accumulation of excessive ROS, which induced severe damage with disruption of physiological processes resulting in retardation of growth (Fig 7). Foliar application of Put significantly reduced $H_2O_2$ and MDA

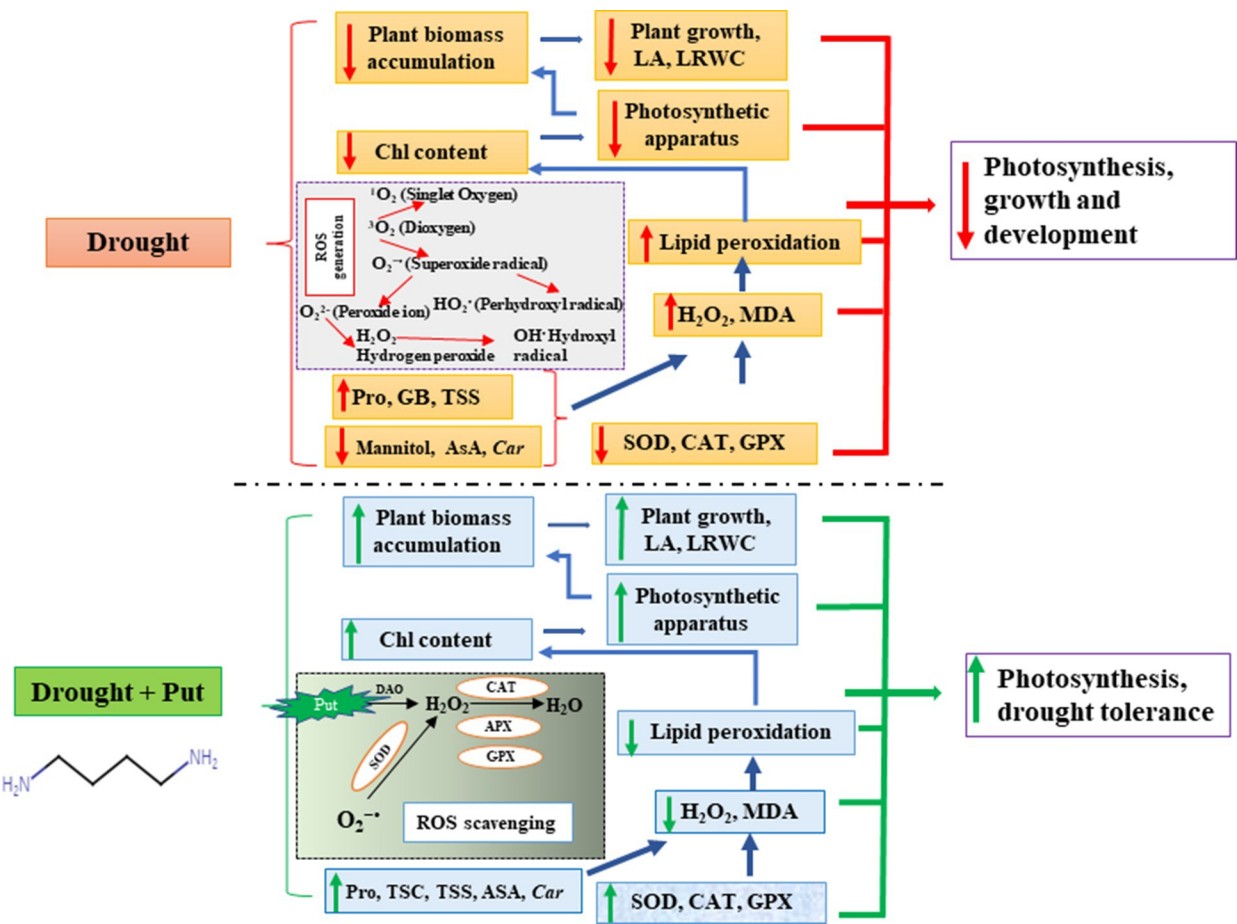

**Fig 7. Schematic representation of drought-induced growth inhibition and its recovery by exogenously Put treatment on sugar beet seedlings.** *Chl*, chlorophyll; *Car*, carotenoid; LA, leaf area; LRWC, leaf relative water content; $H_2O_2$, hydrogen peroxide; ROS, reactive oxygen species; Pro, proline; GB, glycine betaine; TSC, total soluble carbohydrate; TSS, total soluble sugar; AsA, ascorbic acid; SOD, superoxide dismutase; CAT, catalase; APX, ascorbate peroxidase; GPX, guaiacol peroxidase; MDA, malondialdehyde; → and → indicating the effect of drought and putrescine on different variables respectively, → indicating the inter-relationship among the variables.

accumulation by regulating osmoprotectant molecules and antioxidant enzymes. The action of Put helped to protect the photosynthetic pigments, fluorescence, leaf area, and leaf relative water content from drought-stressed seedlings, which maintained higher photosynthesis and other morpho-physiological mechanisms like growth rate and plant dry matter accumulation (Fig 7). Exogenous application of Put enhances drought tolerance of sugar beet seedlings by ROS-scavenging and protecting cells through several morpho-physiological and biochemical processes under drought stress conditions. For most of the traits under study, 0.3 mM putrescine was found effective for both sugar beet genotypes.

## Supporting information

**S1 Fig.**
(DOCX)

**S2 Fig.**
(DOCX)

**S3 Fig.**
(DOCX)

**S4 Fig.**
(DOCX)

**S5 Fig.**
(DOCX)

## Acknowledgments

We thank Dr. Md. Amzad Hossain, Director General, Bangladesh Sugarcrop Research Institute; Dr. Samajit Kumar Pal, Director (Research) and Project Director (SIRF), Bangladesh Sugarcrop Research Institute for supporting this research. We thank Kyoung-Ou Ryu of Asia Seed Co. LTD., Korea, for providing seeds.

## Author Contributions

**Conceptualization:** Md Jahirul Islam, Young-Seok Lim.

**Data curation:** Md Jahirul Islam.

**Formal analysis:** Md Jahirul Islam.

**Investigation:** Md Jahirul Islam, Md Jalal Uddin.

**Methodology:** Md Jahirul Islam.

**Project administration:** Young-Seok Lim.

**Resources:** Juhee Ahn, Eun Ju Cheong, Young-Seok Lim.

**Supervision:** Young-Seok Lim.

**Writing – original draft:** Md Jahirul Islam.

**Writing – review & editing:** Mohammad Anwar Hossain, Robert Henry, Mst. Kohinoor Begum, Md. Abu Taher Sohel, Masuma Akter Mou, Juhee Ahn, Eun Ju Cheong.

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
