## [Decision Letter · Decision Letter 0]

16 Jun 2021

PONE-D-21-13580

Exogenous putrescine attenuates the negative impact of drought stress by modulating physio-biochemical traits and gene expression in sugar beet (Beta vulgaris L.)

PLOS ONE

Dear Prof. Dr. Mohammad Anwar Hossain,

Thank you for submitting your manuscript to PLOS ONE. After careful consideration, we feel that it has merit but does not fully meet PLOS ONE’s publication criteria as it currently stands. Therefore, we invite you to submit a revised version of the manuscript that addresses the points raised during the review process.

Please submit your revised manuscript by July 14, 2021. If you will need more time than this to complete your revisions, please reply to this message or contact the journal office at plosone@plos.org. Please include the following items when submitting your revised manuscript:

We look forward to receiving your revised manuscript.

Kind regards,

Mohammad Golam Mostofa, PhD

Academic Editor

PLOS ONE

Journal Requirements: 

3.Thank you for stating the following in your manuscript: 

This research received a grant from the project entitled ‘Strengthening Integrated Research Facilities of Bangladesh Sugarcane Research Institute’ of Bangladesh Sugarcrop Research Institute (Funding number: 224066000)

4. We note that you have stated that you will provide repository information for your data at acceptance. Should your manuscript be accepted for publication, we will hold it until you provide the relevant accession numbers or DOIs necessary to access your data. If you wish to make changes to your Data Availability statement, please describe these changes in your cover letter and we will update your Data Availability statement to reflect the information you provide

5.PLOS requires an ORCID iD for the corresponding author in Editorial Manager on papers submitted after December 6th, 2016. Please ensure that you have an ORCID iD and that it is validated in Editorial Manager. To do this, go to ‘Update my Information’ (in the upper left-hand corner of the main menu), and click on the Fetch/Validate link next to the ORCID field. This will take you to the ORCID site and allow you to create a new iD or authenticate a pre-existing iD in Editorial Manager. Please see the following video for instructions on linking an ORCID iD to your Editorial Manager account: https://www.youtube.com/watch?v=_xcclfuvtxQ

Reviewers' comments:

Reviewer's Responses to Questions

**Comments to the Author**

1. Is the manuscript technically sound, and do the data support the conclusions?

Reviewer #1: Yes

Reviewer #2: Yes

2. Has the statistical analysis been performed appropriately and rigorously? 

Reviewer #1: Yes

Reviewer #2: Yes

3. Have the authors made all data underlying the findings in their manuscript fully available?

Reviewer #1: Yes

Reviewer #2: Yes

4. Is the manuscript presented in an intelligible fashion and written in standard English?

Reviewer #1: No

Reviewer #2: Yes

5. Review Comments to the Author

Reviewer #1: Authors have very well showed that application of putrescine (Put) alleviates drought stress in two sugar beet genotypes. Authors have studied effect of Put application in sugar beets during drought stress in various biological processes such as physiological, biochemical, and transcriptional level. Moreover, Authors have demonstrated that Put application enhances antioxidant levels thereby reduces H2O2 and MDA level during drought stress in Sugar beet genotypes.

However, there are several formatting errors and in detail explanation needed in material method section and in every figure legend.

There are several grammatical errors in spacing, manuscript needs to be thoroughly checked for spacing errors.

Why it has been not considered to treat control plants also with Putrescine and normalize the data? Furthermore, why authors have not showed any images of plants for example how sugar beet plants looks after 10 days of drought stress and how Put treatment recovers severity of drought phenotype?

Introduction:

Line no. 67-68; reference error need to be checked.

Spacing errors.

Material and methods:

Line no. 115 to 117; explain in detail for how long (time) seeds were sterilized.

Line 121 to 125; This is the important information for readers, authors should mention whether Put was sprayed for every day/how many times per day?

Line 176: reference error, numbering not mentioned to reference.

Result and Discussion:

Result section is hard to follow. Authors should consider to rewrite it.

Line 226 to 236: written very poorly, Authors should consider rewriting this section carefully, What is Kaveri? The table 2 is very confusing, there are no units to Shoot dry wt, root dry wt, leaf area, LRWC. Again, new Kaveri genotype which is not mentioned in materials and method section!

Plant growth and parameters: At which day samples were collected/analyzed not mentioned.

At line 247 to 250, there is no flow to the sentence.

Line 302 to 306, there is no flow while reading, consider rewriting.

Antioxidant Enzyme Activity:

Line 405 to 406: Written wrong! Antioxidant enzyme levels were reduced after drought in both genotypes but its written increased; though its being increased in Put treatment compared to drought; but this has been not mentioned. The results written in Antioxidant enzyme activity need to rewrite carefully.

Relative gene expression:

If the relative gene expression is normalized to reference gene, then it should be mentioned and if it is compared to something then it is fold change or Log2fold change?

Authors need to consider checking the expression of Put biosynthesis genes namely, arginine decarboxylase (ADC) and Orn decarboxylase (ODC) in their sugar beet samples. Authors have checked expression of antioxidant genes after 10 days of drought, earlier time point is important. Moreover, whether does Put treatment to control sugar beet plants/genotypes alters the expression of these antioxidant genes has not been verified, short experiment would be to just treat sugar beet seedlings with Put for several timepoints and check expression of these antioxidant genes along with Put biosynthesis genes.

Figures:

Figure legends must be rewritten! In all figures, It is very confusing to determine the significant differences between the treatments, Authors should consider labeling significant differences by different letters and non-significance by same letter. This way it is easy for readers to follow. Moreover, Authors should explain in figure legend about the asterisk/ letters (if newly added) denotes significant differences between which treatment.

Reviewer #2: In the manuscript (ms) entitled “Exogenous putrescine attenuates the negative impact of drought stress by modulating physio-biochemical traits and gene expression in sugar beet (Beta vulgaris L.)” authors put quite a lot effort. Here they investigated the potential roles of exogenous putrescine (Put) in improving drought stress tolerance of two sugar beet genotypes by assessing morpho-physiological and biochemical traits, as well as gene expression.

In my opinion, the findings of this study would be pretty interesting and important for the readers from related field. However, the ms in the current state lacks cohesiveness in scientific writing and prepared carelessly in terms of writing norm. Based on the following observations the article in its current status is not suitable to publish in this journal.

1. In the abstract, the authors mainly discussed the methods and listed the overall results instead of presenting the main findings. Authors are suggested to synthesize the main findings here highlighting special techniques/methods used.

2. The main drawback of this ms is poor presentation and explanation of results. In presenting comparative results, authors in most of the cases just have used the terms ‘significant’, ‘prominent increase’, ‘increased/decreased’. In this case, authors should focus on quantifying (where possible) the recovery of drought-induced losses in sugar beet in response to Put treatments. Additionally, authors should include a comparative phenotype figure on Put-induced drought tolerance in sugar beet to strengthen their claim.

3. The subsequent discussion supporting the results has also been very superficial. For instance, in Line (L)-270, authors mentioned “these results have been confirmed in other studies”. How their results confirmed by others? Authors should clearly express what they meant here?

4. The ms requires several rounds of English polishing. Specifically, there are lots of typos, long sentences and scientifically incorrect sentences in the ms, which might confuse the readers to get clear insights. Inconsistency in writing style (abbreviation, citation etc.) should also address very seriously.

Some specific concerns, but by no means an exhaustive list, mentioned below should be addressed to improve the current status of the ms:

-L67: Citation style is found inconsistent

-while discussing the results in L225: surprisingly, in table 2, no ‘'BSRI Sugar beet 2’ was found!! Here they mentioned the genotype name 'Kaveri' instead of 'BSRI Sugar beet 2'. It seems authors carelessly mentioned ‘Kaveri’ here!! Or, something else, please clarify.

-L231-232: “In the case of SBT-010, although SDW and LA recovered

most under 0.6 mM and 0.9 mM Put, respectively, but no positive effect was observed for RDW”. As said earlier, from such types of ‘English’ readers might lose their interest. Please improve the English standard to have readers’ interest.

L258: chlorophyll "a" and "b" should write in italic!

L280: Results explained on 'fluorescence parameters' is not understandable at all!

L281: What is meant by OJIP? Author should elaborately mention when any abbreviation appears first.

L294-296: Unnecessary discussion.

L405-411: Very hard to understand the activity of enzymes. This explanation is not even consistent with Fig. 4!

L469: Authors mentioned ‘it was revealed that cluster-A was mostly characterized by…….’ They should avoid such kind of vague and non-scientific wording to present the data.

L485-491: Authors should discuss the PCA results more elaborately to clearly focus the main traits contributed greatly to Put-mediated drought tolerance in sugar beet.

L519-520: In the last line of conclusion “Importantly, Put at a medium dose (0.6 mM) was found to be most effective in modulating the morpho-physiological and biochemical processes in sugar beet plants under drought stress. I wonder how authors reached on a conclusion like this! Besides, the term ‘medium dose’ is a vague and non-scientific word!

-the figure and table legends are not complete. Authors should give more care on writing it completely.

-in table 4: in presenting the Pro data of SBT-010, it seems authors mistakenly put the same letter ‘g’ for both ‘4.81 ± 0.48’ and ‘15.05 ± 0.07’. There must have significant difference between them! Authors should seriously address such kind of issue.

Further recommendation

It would be pretty interesting and worth reading for the related readers if authors could produce a flow/process diagram on Put-mediated drought tolerance mechanism in sugar beet based on the findings of the present study.

6. PLOS authors have the option to publish the peer review history of their article (what does this mean?). If published, this will include your full peer review and any attached files.

Reviewer #1: No

Reviewer #2: **Yes: **Gopal Saha

---

## [Author Response · Author response to Decision Letter 0]

30 Jul 2021

Reviewer 1: I have incorporated your suggestions into my revision. They were very helpful. Thank you.

Reviewer 2: I have incorporated your suggestions into my revision. They were very helpful. Thank you.

---

## [Decision Letter · Decision Letter 1]

31 Aug 2021

PONE-D-21-13580R1

Exogenous putrescine attenuates the negative impact of drought stress by modulating physio-biochemical traits and gene expression in sugar beet (Beta vulgaris L.)

PLOS ONE

Dear Dr. Hossain,

Thank you for submitting your manuscript to PLOS ONE. After careful consideration, we feel that it has merit but does not fully meet PLOS ONE’s publication criteria as it currently stands. Therefore, we invite you to submit a revised version of the manuscript that addresses the points raised during the review process.

We look forward to receiving your revised manuscript.

Kind regards,

Mohammad Golam Mostofa, PhD

Academic Editor

PLOS ONE

Journal Requirements:

Additional Editor Comments (if provided):

Authors should present the figures by placing the data of two varieties side by side in one figure for better comparison. Authors are also suggested to use either SEs or SDs in the tables and figures. There are still many English issues throughout the manuscript.

Reviewers' comments:

Reviewer's Responses to Questions

**Comments to the Author**

1. If the authors have adequately addressed your comments raised in a previous round of review and you feel that this manuscript is now acceptable for publication, you may indicate that here to bypass the “Comments to the Author” section, enter your conflict of interest statement in the “Confidential to Editor” section, and submit your "Accept" recommendation.

Reviewer #1: All comments have been addressed

Reviewer #2: All comments have been addressed

2. Is the manuscript technically sound, and do the data support the conclusions?

Reviewer #1: Yes

Reviewer #2: Partly

3. Has the statistical analysis been performed appropriately and rigorously? 

Reviewer #1: Yes

Reviewer #2: No

4. Have the authors made all data underlying the findings in their manuscript fully available?

Reviewer #1: Yes

Reviewer #2: Yes

5. Is the manuscript presented in an intelligible fashion and written in standard English?

Reviewer #1: Yes

Reviewer #2: Yes

6. Review Comments to the Author

Reviewer #1: Dear Authors,

If possible enlarge the newly added plant photographs for better visualization.

Please check the manuscript once again for grammatical mistakes, I see at some places there are spacing errors.

Thank you.

Reviewer #2: I thank Islam et al. for their effort in revising and updating the status of the manuscript (ms). In the revised ms authors remarkably improved the results writing and English standard.

However, I have a major concern on the current data presentation style both in tables and figures. I am surprised why authors presented data (table 2-4) in separate panels for two different varieties. Authors must justify this issue first. To me, it would have been better to present data in one panel after running a two-way ANOVA. This will provide detailed information on main and interactive effects. Therefore, the reader will be able to assess the treatments across different concentrations of Putrescine (Put) and variety. I suggest the authors to consult with a professional statistician to get better insight about the analysis. Likewise, for better visibility of the treatment effects in figures, authors are recommended to present the data of two varieties combinedly as single comparative group bar graph (for each parameter) instead of two separate figures.

Besides, there are some other concerns listed below and authors should address these before publication of the ms:

L126-127(Materials & Methods)-Authors should tell about the commercial formulation of Put, e. g., form, company name etc. they have been used for treatment.

L454-478 (Results & discussion)- The Relative gene expression section has not been well written as per data presented in the fig. 5. To me, in most of the cases the relative expression of antioxidant genes contrasts between the two varieties in response to Put treatments!! Authors should discuss the results accordingly and focus this issue with possible facts.

L563-564 (Conclusion)-the authors concluded “However considering most of the traits under study, 0.3 mM TU was of effective for the variety BSRI Sugar beet 2 whereas 0.6 mM TU was found was found effective for the variety SBT-010” – what actually authors meant by ‘TU’? Importantly, how authors recommended two different doses of Put are suitable for drought tolerance of two sugar beet varieties, since authors analyzed and presented the interaction data separately?? In that case authors may specifically mention the traits/parameters of two varieties improved for drought tolerance in response to Put application.

Finally, the figures of phenotypic differences of treatment effects were supposed to be presented in a separate figure, authors provided it along with the schematic diagram in a less convincing way! However, for better understanding for the readers the authors should give the discussion on this Put-induced drought tolerance mechanism (fig. 7) separately as individual section other than depicting it only in the figure legend! And, authors are advised to present this figure in a more reader friendly way explaining all the items (e.g., arrow, color code etc.) of the figure in the figure legend.

7. PLOS authors have the option to publish the peer review history of their article (what does this mean?). If published, this will include your full peer review and any attached files.

Reviewer #1: No

Reviewer #2: **Yes: **Gopal Saha

---

## [Author Response · Author response to Decision Letter 1]

27 Oct 2021

Comments of Reviewer #1: 

Dear Authors,

If possible enlarge the newly added plant photographs for better visualization.

Please check the manuscript once again for grammatical mistakes, I see at some places there are spacing errors.

Thank you.

Author’s response: Thank you very much for your valuable suggestions which helped us to improve the manuscript. A new figure (Fig. 3C) depicting the phenotypic differences of both sugar beet genotypes under control, drought, and D + 0.3 mM Put (best treatment) has been added in the revised manuscript for better understanding of the readers. 

Comments of Reviewer #2: 

I thank Islam et al. for their effort in revising and updating the status of the manuscript (ms). In the revised ms authors remarkably improved the results writing and English standard.

However, I have a major concern on the current data presentation style both in tables and figures. I am surprised why authors presented data (table 2-4) in separate panels for two different varieties. Authors must justify this issue first. To me, it would have been better to present data in one panel after running a two-way ANOVA. This will provide detailed information on main and interactive effects. Therefore, the reader will be able to assess the treatments across different concentrations of Putrescine (Put) and variety. I suggest the authors to consult with a professional statistician to get better insight about the analysis. Likewise, for better visibility of the treatment effects in figures, authors are recommended to present the data of two varieties combinedly as single comparative group bar graph (for each parameter) instead of two separate figures.

Author’s response: Thank you very much for your valuable comments and suggestions once again. We revised all the tables and graphs where tables represented in one panel after running a two-way ANOVA and all figures represented by combinedly as a single comparative group bar graph for two genotypes for easy understanding of the readers.

Besides, there are some other concerns listed below and authors should address these before publication of the ms:

L126-127(Materials & Methods)-Authors should tell about the commercial formulation of Put, e. g., form, company name etc. they have been used for treatment.

Author’s response: The commercial formulation of Putrescine and manufacturer information has been added in the materials and methods section accordingly (line 126-127).

L454-478 (Results & discussion)- The Relative gene expression section has not been well written as per data presented in the fig. 5. To me, in most of the cases the relative expression of antioxidant genes contrasts between the two varieties in response to Put treatments!! Authors should discuss the results accordingly and focus this issue with possible facts.

Author’s response: The relative gene expression section has been amended accordingly.

L563-564 (Conclusion)-the authors concluded “However considering most of the traits under study, 0.3 mM TU was of effective for the variety BSRI Sugar beet 2 whereas 0.6 mM TU was found was found effective for the variety SBT-010” – what actually authors meant by ‘TU’? Importantly, how authors recommended two different doses of Put are suitable for drought tolerance of two sugar beet varieties, since authors analyzed and presented the interaction data separately?? In that case authors may specifically mention the traits/parameters of two varieties improved for drought tolerance in response to Put application.

Author’s response: We apologize for our mistake in writing the effective dose of putrescine in the conclusion section. In the revised manuscript, we recommended single dose of Put based on the two-way ANOVA analysis as your suggested.

Finally, the figures of phenotypic differences of treatment effects were supposed to be presented in a separate figure, authors provided it along with the schematic diagram in a less convincing way! However, for better understanding for the readers the authors should give the discussion on this Put-induced drought tolerance mechanism (fig. 7) separately as individual section other than depicting it only in the figure legend! And, authors are advised to present this figure in a more reader friendly way explaining all the items (e.g., arrow, color code etc.) of the figure in the figure legend.

Author’s response: A new figure (Fig. 3C) depicting the phenotypic differences of both sugar beet genotypes under control, drought, and D + 0.3 mM Put (best treatment) has been added in the revised manuscript for better understanding of the readers. The discussion on Put-induced drought tolerance mechanism (Fig. 7) has been added separately as you suggested (line 545-554). The explanation related to the arrow and color code was also added in the revised figure legend accordingly.

Comments of academic editor:

Authors should present the figures by placing the data of two varieties side by side in one figure for better comparison. Authors are also suggested to use either SEs or SDs in the tables and figures. There are still many English issues throughout the manuscript.

Author’s response: Thanks for your meaningful comments. We have revised our MS based on your comments and reviewer’s comments. A new figure (Fig. 3C) depicting the phenotypic differences of both sugar beet genotypes under control, drought, and D + 0.3 mM Put (best treatment) has been added in the revised manuscript for better understanding of the readers. SD values were also added in the figures and tables. The english of the MS was futher carefully checked by Professor Robert Henry.

---

## [Decision Letter · Decision Letter 2]

19 Dec 2021

Exogenous putrescine attenuates the negative impact of drought stress by modulating physio-biochemical traits and gene expression in sugar beet (Beta vulgaris L.)

PONE-D-21-13580R2

Dear Dr. Hossain,

We’re pleased to inform you that your manuscript has been judged scientifically suitable for publication and will be formally accepted for publication once it meets all outstanding technical requirements.

Kind regards,

Mohammad Golam Mostofa, PhD

Academic Editor

PLOS ONE

Additional Editor Comments (optional):

Please address the minor issues raised by Reviewer 2 during proof reading.

Reviewers' comments:

Reviewer's Responses to Questions

**Comments to the Author**

1. If the authors have adequately addressed your comments raised in a previous round of review and you feel that this manuscript is now acceptable for publication, you may indicate that here to bypass the “Comments to the Author” section, enter your conflict of interest statement in the “Confidential to Editor” section, and submit your "Accept" recommendation.

Reviewer #2: All comments have been addressed

2. Is the manuscript technically sound, and do the data support the conclusions?

Reviewer #2: Yes

3. Has the statistical analysis been performed appropriately and rigorously? 

Reviewer #2: Yes

4. Have the authors made all data underlying the findings in their manuscript fully available?

Reviewer #2: Yes

5. Is the manuscript presented in an intelligible fashion and written in standard English?

Reviewer #2: Yes

6. Review Comments to the Author

Reviewer #2: I thank Islam et al. for their effort in revising and updating the status of the manuscript (ms). In the revised ms authors remarkably addressed the major concern on statistical issues and data presentation style, especially the tables and figures.

Before publication authors should address the following issues for better readability of the article:

For all figures (1-6): the lettering on the top of the bar should be positioned nicely comparing with other font size in fig.

Fig 3: Why the comparative phenotypic effect of D+0.6 mM Put & D+0.9 mM Put have not been presented? Please add these if they are available!

Fig 5: Please use contrasting color code for bars instead of current format for better understanding. The scaling font in the X & Y axis looks unusually large!

Fig 6: In the PCA plot (both 1&2) the variables/treatments names are overlapped due to space problem. Authors may consider customizing the writing/presenting style either using symbol, or omitting them and re-writing in distant understandable format & font.

7. PLOS authors have the option to publish the peer review history of their article (what does this mean?). If published, this will include your full peer review and any attached files.

Reviewer #2: **Yes: **Gopal Saha

---

## [Editor Report · Acceptance letter]

26 Dec 2021

PONE-D-21-13580R2 

Exogenous putrescine attenuates the negative impact of drought stress by modulating physio-biochemical traits and gene expression in sugar beet (*Beta vulgaris* L.) 

Dear Dr. Hossain:

I'm pleased to inform you that your manuscript has been deemed suitable for publication in PLOS ONE. Congratulations! Your manuscript is now with our production department. 

Kind regards, 

on behalf of

Dr. Mohammad Golam Mostofa 

Academic Editor

PLOS ONE